# iTAP, a novel iRhom interactor, controls TNF secretion by policing the stability of iRhom/TACE

Ioanna Oikonomidi[1], Emma Burbridge[1], Miguel Cavadas[1], Graeme Sullivan[2], Blanka Collis[3], Heike Naegele[4], Danielle Clancy[2], Jana Brezinova[3], Tianyi Hu[1], Andrea Bileck[5†‡], Christopher Gerner[5], Alfonso Bolado[6], Alex von Kriegsheim[6], Seamus J Martin[2], Florian Steinberg[4], Kvido Strisovsky[3], Colin Adrain[1]*

[1]Membrane Traffic Lab, Instituto Gulbenkian de Ciência, Oeiras, Portugal; [2]Molecular Cell Biology Laboratory, Department of Genetics, The Smurfit Institute, Trinity College Dublin, Dublin, Ireland; [3]Institute of Organic Chemistry and Biochemistry, Academy of Sciences of the Czech Republic, Prague, Czech Republic; [4]Center for Biological Systems Analysis, Faculty of Biology, Albert Ludwigs Universitaet Freiburg, Freiburg, Germany; [5]Institut für Analytische Chemie, Universität Wien, Vienna, Austria; [6]Edinburgh Cancer Research UK Centre, Institute of Genetics and Molecular Medicine, University of Edinburgh, Edinburgh, United Kingdom

*For correspondence:
cadrain@igc.gulbenkian.pt

Present address: †Department of Clinical Research, Bern University Hospital, Bern, Switzerland; ‡Department of Nephrology and Hypertension, Bern University Hospital, Bern, Switzerland

Competing interests: The authors declare that no competing interests exist.

**Abstract** The apical inflammatory cytokine TNF regulates numerous important biological processes including inflammation and cell death, and drives inflammatory diseases. TNF secretion requires TACE (also called ADAM17), which cleaves TNF from its transmembrane tether. The trafficking of TACE to the cell surface, and stimulation of its proteolytic activity, depends on membrane proteins, called iRhoms. To delineate how the TNF/TACE/iRhom axis is regulated, we performed an immunoprecipitation/mass spectrometry screen to identify iRhom-binding proteins. This identified a novel protein, that we name iTAP (iRhom Tail-Associated Protein) that binds to iRhoms, enhancing the cell surface stability of iRhoms and TACE, preventing their degradation in lysosomes. Depleting iTAP in primary human macrophages profoundly impaired TNF production and tissues from iTAP KO mice exhibit a pronounced depletion in active TACE levels. Our work identifies iTAP as a physiological regulator of TNF signalling and a novel target for the control of inflammation.

DOI: https://doi.org/10.7554/eLife.35032.001

## Introduction

The cytokine TNF controls numerous important biological processes (e.g. inflammation, fever, apoptosis, necroptosis, cachexia, tumorigenesis, viral infection, insulin signaling) and is heavily implicated in inflammatory disease (*Brenner et al., 2015*). Anti-TNF biologics are the highest-selling drugs internationally and there is intense interest in how TNF signaling is regulated (*Brenner et al., 2015*). TNF is expressed by a range of cells including macrophages, lymphocytes, natural killer cells, endothelial cells and microglia and is synthesized as a type II transmembrane protein with a cytosolic domain of 76 amino acids that assembles into a trimer (*Locksley et al., 2001*). The capacity of TNF to trigger such pleiotropic biological outcomes is determined by its ability to activate two distinct receptors (*Locksley et al., 2001*). Generally, TNFRI activation is associated with induction of acute or chronic inflammatory responses, or cell death, whereas TNFRII mediates pro-survival signals and

**eLife digest** Inflammation forms part of the body's defense system against pathogens, but if the system becomes faulty, it can cause problems linked to inflammatory and autoimmune diseases. Immune cells coordinate their activity using specific signaling molecules called cytokines. For example, the cytokine TNF is an important trigger of inflammation and is produced at the surface of immune cells. A specific enzyme called TACE is needed to release TNF, as well as other signaling molecules, including proteins that trigger healing.

Previous work revealed that TACE works with proteins called iRhoms, which regulate its activity and help TACE to reach the surface of the cell to release TNF. To find out how, Oikonomidi et al. screened human cells to see what other proteins interact with iRhoms. The results revealed a new protein named iTAP, which is required to release TNF from the surface of cells. It also protects the TACE-iRhom complex from being destroyed by the cell's waste disposal system.

When iTAP was experimentally removed in human immune cells, the cells were unable to release TNF. Instead, iRhom and TACE travelled to the cell's garbage system, the lysosome, where the proteins were destroyed. Removing the iTAP gene in mice had the same effect, and the TACE-iRhom complex was no longer found on the surface of the cell, but instead degraded in lysosomes. This suggests that in healthy cells, the iTAP protein prevents the cell from destroying this protein complex.

TNF controls many beneficial processes, including fighting infection and cancer. However, when the immune system releases too many cytokines, it can lead to inflammatory diseases or even cause cancer. Specific drugs that target TNF are not always effective administered on their own, and sometimes, patients stop responding to the drugs. Since the new protein iTAP works as a switch to turn TNF release on or off, it could provide a target for the development of new treatments.

DOI: https://doi.org/10.7554/eLife.35032.002

has been associated with the tolerogenic properties of regulatory T cells (*Kleijwegt et al., 2010*; *Richter et al., 2012*).

Additional complexity is imposed by the form of TNF that engages in signalling. The transmembrane form of TNF (mTNF) triggers juxtacrine signalling, while TNF released as a soluble form (sTNF), drives paracrine signalling. Notably, whereas TNFRI can be activated by both soluble and membrane TNF, TNFRII is activated efficiently only by mTNF (*Grell et al., 1995*). Whereas mTNF is sufficient for the development and maintenance of some lymphoid tissues, soluble TNF is important for acute and chronic inflammation. Specifically, mice unable to produce soluble TNF resist endotoxic shock and exhibit reduced severity in experimental autoimmune encephalomyelitis models (*Ruuls et al., 2001*). Membrane TNF is also insufficient to rescue the defects in primary B cell follicle formation observed in TNF KO mice, but mediates protective immune responses to intracellular bacterial infection (*Ruuls et al., 2001*; *Alexopoulou et al., 2006*).

The ability to engage biological outcomes that require TNFRI versus TNFRII, (and the capacity to control the physical distance over which signaling is effective) therefore, critically depends on the ability to release soluble TNF from the cell surface. This is catalyzed by the protease TACE (TNF α Converting Enzyme) (*Horiuchi et al., 2007*; *Peschon et al., 1998*), also called ADAM17 (A Disintegrin And Metalloprotease) (*Gooz, 2010*; *Zunke and Rose-John, 2017*). Crucially, TACE imposes an additional layer of versatility and regulation to TNF signaling, since in addition to cleaving TNF, both TNFRs are also physiological TACE substrates. Hence, TACE is a master orchestrator of TNF signaling, tuning signaling to fit a panoply of biological roles ranging from inflammatory responses to immune tolerance. TACE also has significant biological importance beyond TNF signaling since it cleaves other prominent substrates, including the activating ligands of the EGFR (Epidermal Growth Factor Receptor), an important pathway that drives growth control, tissue repair and immune responses.

Given its ability to elicit potent biological responses, it is unsurprising that TACE is stringently regulated (*Murphy, 2009*; *Grötzinger et al., 2017*). A major control point in TACE regulation involves its trafficking within the secretory pathway (*Schlöndorff et al., 2000*). TACE is synthesized in the endoplasmic reticulum (ER) as a catalytically inactive precursor. For TACE to become

proteolytically active, it must undergo a maturation step—removal of its prodomain—which is catalysed by proprotein convertases in the *trans*-Golgi (*Schlöndorff et al., 2000*). The exit of TACE from the ER and its trafficking to the cell surface requires regulatory proteins called iRhoms (*Adrain et al., 2012*; *McIlwain et al., 2012*; *Li et al., 2015*). Hence, iRhom KO mice, or cells in which iRhoms are ablated, lack TACE activity (*Adrain et al., 2012*; *Li et al., 2015*; *Christova et al., 2013*). Mice null for iRhom2, whose expression is enriched in myeloid cells, cannot secrete TNF (*Adrain et al., 2012*; *McIlwain et al., 2012*; *Siggs et al., 2012*).

An important checkpoint to license TACE activity involves its stimulation on the cell surface by agents including phorbol esters, Toll-like receptor agonists and G-protein coupled receptor ligands (*Grötzinger et al., 2017*; *Arribas et al., 1996*; *Hall and Blobel, 2012*; *Brandl et al., 2010*; *Wetzker and Böhmer, 2003*). Importantly, as well as controlling TACE trafficking, iRhoms exist in a molecular assembly with TACE on the cell surface—the 'sheddase complex' which is central to stimulation of TACE sheddase activity. Within the sheddase complex, iRhom proteins serve as a platform that senses and transduces TACE-activating stimuli. These agents provoke the MAP kinase-dependent phosphorylation of the iRhom2 cytoplasmic tail, which in turn triggers the recruitment of 14-3-3 proteins. This enforces the detachment of TACE from iRhom2 (or triggers a conformational change within the sheddase complex) which is required to facilitate TACE's ability to cleave its substrates, including TNF (*Grieve et al., 2017*; *Cavadas et al., 2017*). Hence, iRhoms are allosteric regulators of TACE's proteolytic activity as well as acting as trafficking factors.

As iRhom and TACE form an intrinsic complex, their trafficking itinerary and fate within the secretory pathway must presumably be interdependent. In spite of the importance of trafficking for TACE regulation (*Schlöndorff et al., 2000*; *Adrain et al., 2012*; *Dombernowsky et al., 2015*), surprisingly little is known about the machinery that controls TACE, or iRhom, trafficking to/from the plasma membrane. An exception is PACS-2 (Phosphofurin Acidic Cluster Sorting Protein 2), a protein that colocalizes with mature TACE in endocytic compartments (*Dombernowsky et al., 2015*) and controls its endocytic recycling. Ablation of PACS-2 in cells impairs the cell surface availability of TACE, reducing substrate cleavage (*Dombernowsky et al., 2015*). However, PACS-2 has a relatively modest impact on TACE biology *in vivo* (*Dombernowsky et al., 2017*), suggesting the possibility of unidentified trafficking regulators that may act separately from, or redundantly with, PACS-2.

As iRhoms form functionally important complexes with cell surface TACE (*Grieve et al., 2017*; *Cavadas et al., 2017*; *Maney et al., 2015*), modulation of iRhom trafficking in the endocytic pathway has the potential to act as a regulatory mechanism that controls TNF secretion. It has been shown that not only TACE (*Doedens and Black, 2000*; *Lorenzen et al., 2016*), but also iRhoms (*Grieve et al., 2017*; *Cavadas et al., 2017*) are endocytosed and degraded in lysosomes, but the machinery involved in maintaining stable cell surface levels of the sheddase complex is unknown.

Here we identify a novel protein that we name iTAP (iRhom Tail-Associated Protein) that is essential for the control of the stability of iRhom2 and TACE on the plasma membrane. Ablation of iTAP triggers the mis-sorting of iRhom2, and consequently, TACE, to lysosomes, where they are degraded. Consistent with this, loss of iTAP results in a dramatic reduction in TACE activity and TNF secretion. Our work reveals iTAP as a key physiological regulator of TNF release.

## Results

### iTAP, a novel interactor of iRhoms, is an atypical FERM domain-containing protein

To identify novel regulators of mammalian iRhoms 1 and −2, we adopted an immunoprecipitation/mass spectrometry (IP/MS) approach described in our previous work (*Cavadas et al., 2017*). As shown in *Figure 1A*, we generated a panel of HEK 293ET cell lines stably expressing HA-tagged forms of full-length iRhom1, iRhom2, or the iRhom1 N-terminal cytoplasmic tail only. To focus only on proteins that bind selectively to iRhoms, we included the related rhomboid-like proteins, Rhbdd2, RHBDD3, Ubac2, as specificity controls (*Figure 1A*). As expected, only immunoprecipitates (IPs) from cells expressing full-length HA-tagged iRhom1 or iRhom2 captured endogenous TACE, confirming the validity of the approach (*Figure 1B*).

To identify novel iRhom-binding proteins, we subjected IPs from these cells to mass spectrometry. This analysis revealed, in multiple replicate experiments, peptides from a largely uncharacterized

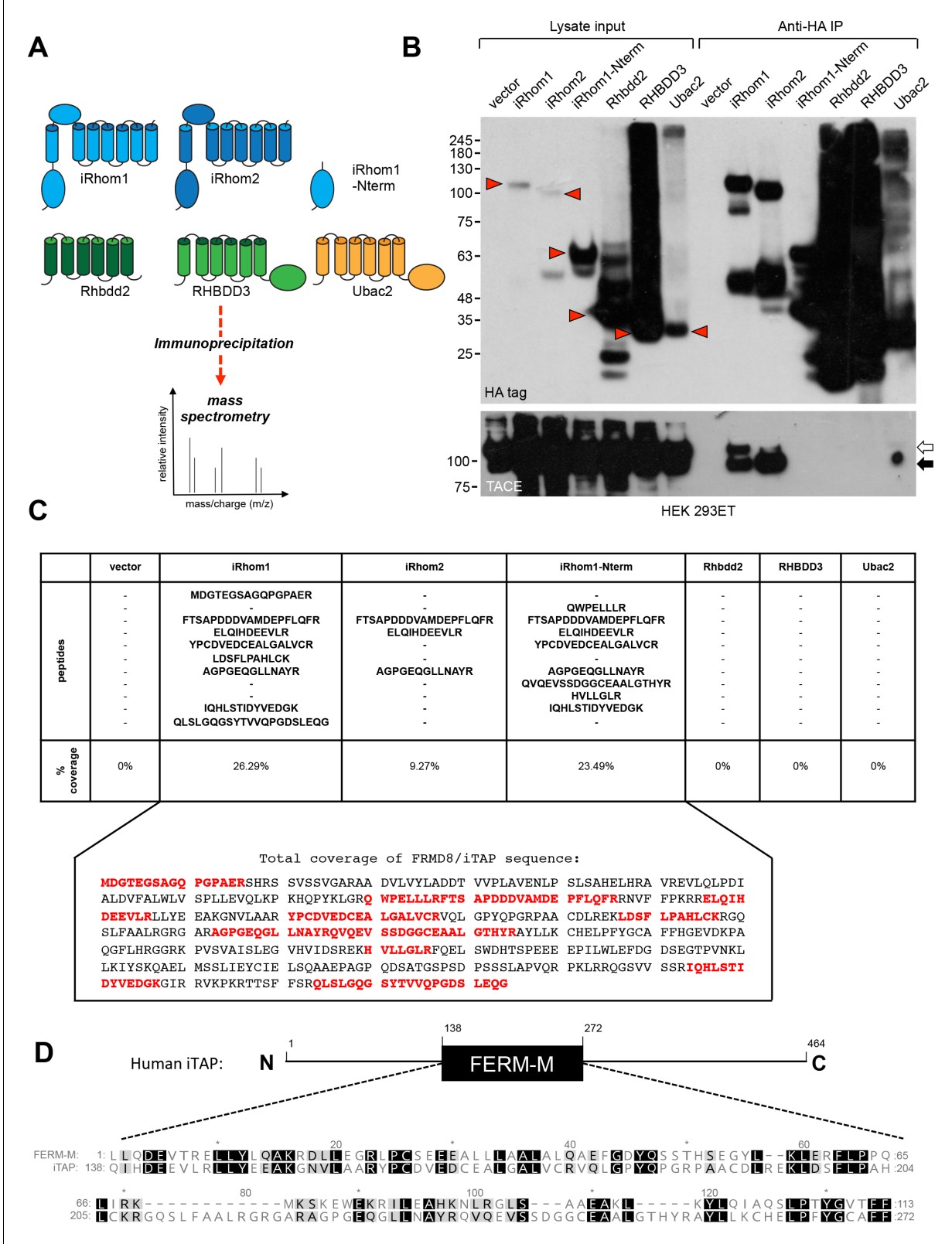

**Figure 1.** Identification of iTAP as a novel iRhom-interacting protein. (**A**). Schematic diagram showing the stable HEK 293ET cell lines expressing iRhom proteins or related rhomboid pseudoproteases as controls, which were subjected to immunoprecipitation followed by mass spectrometry. (**B**). An example immunoprecipitation indicating that only immunoprecipitates from cell lines expressing WT iRhom1 or iRhom2 contain the binding of the positive control protein, TACE. Here and throughout, immature and mature species of TACE are indicated with white and black arrows respectively.

*Figure 1 continued on next page*

*Figure 1 continued*

The red arrowheads show the full-length forms of the individual rhomboid-like proteins. (C). Peptides identified by mass spec that were assigned to FRMD8/iTAP. These peptides were found in immunoprecipitates from iRhom1, iRhom2 or the N-terminus of iRhom1 but not in the other samples. The peptides are shown from a representative experiment. All of the peptides found in the iRhom immunoprecipitates were mapped onto the human FRMD8 amino acid sequence (indicated in red). (D). Schematic diagram illustrating the domain structure of human FRMD8/iTAP. Beneath is shown a CLUSTALW alignment of the pfam FERM-M consensus (pfam00373) alongside FRMD8/iTAP. Identical residues are shaded in black and similar residues in grey.

DOI: https://doi.org/10.7554/eLife.35032.003

The following figure supplement is available for figure 1:

**Figure supplement 1.** iTAP is broadly expressed in a variety of tissues important for TACE biology.

DOI: https://doi.org/10.7554/eLife.35032.004

protein, called FRMD8 (FERM Domain-containing protein 8), in IPs of iRhom1 or iRhom2, but not in control IPs of Rhbdd2, RHBDD3 and Ubac2 (*Figure 1C*). Furthermore, FRMD8 was found in IPs from cells expressing the N-terminus of iRhom1 (*Figure 1A,C*), suggesting that it was recruited to the iRhom cytoplasmic tail (R-domain), an important regulatory region (*Grieve et al., 2017*; *Cavadas et al., 2017*; *Maney et al., 2015*; *Hosur et al., 2014*). In light of this, we named the novel protein iTAP ('iRhom Tail Associated Protein'). A closer inspection of the iTAP sequence revealed that it encodes a FERM (band 4.1/Ezrin/Radixin/Moesin) domain (*Chishti et al., 1998*) (*Figure 1D*).

Proteins containing FERM domains fulfil many important roles, including signaling, organization of the cell cortex and its mechanical properties and cell surface stabilization (anchoring) of membrane proteins or phospholipids (*McClatchey, 2014*; *Fehon et al., 2010*; *Hoover and Bryant, 2000*; *Baines et al., 2014*; *Moleirinho et al., 2013*). The well-characterized FERM domain contains three distinct lobes that together resemble a three-leaf clover (*Pearson et al., 2000*). However, unlike most FERM domain-containing proteins, but similar to its paralog KRIT1—an adaptor protein in the cerebral cavernous malformation pathway that regulates the establishment of vasculature (*Pal et al., 2017*), iTAP contains only the central (FERM-M) lobe (*Figure 1D*). iTAP orthologs are present in metazoans, including *Drosophila* and *Danio* (*Kategaya et al., 2009*). The iTAP protein is expressed broadly and is co-expressed with TACE and iRhom1 or iRhom2 in a range of tissues relevant for TACE biology (*Figure 1—figure supplement 1A,B*).

Independent immunoprecipitation experiments verified that iTAP binds specifically to both iRhom1 and iRhom2, but not to the related rhomboid pseudoproteases Ubac2 and Rhbdd2 (*Figure 2A*). iTAP binds the cytoplasmic tail of iRhoms, since a mutant containing only the cytoplasmic tail of iRhom1 bound iTAP robustly, whereas a mutant lacking all of the iRhom2 cytoplasmic tail (ΔNterm) failed to bind (*Figure 2A*). By contrast, removal of the iRhom homology domain (IRHD), the luminal globular domain between transmembrane helices 1 and 2 (*Figure 2B*) from iRhom2 had no impact on iTAP recruitment (*Figure 2A*). These data indicate that iTAP is specifically recruited to the cytoplasmic tail of iRhoms. Notably, when iTAP-FLAG, but not a panel of control proteins, was immunoprecipitated from cell lysates, we detected the binding of endogenous iRhom1 and iRhom2 to iTAP (*Table 1*). These data confirm that iTAP is a specific endogenous interactor of both iRhom paralogs in mammals.

To delineate the region within the cytoplasmic tail of iRhom that iTAP binds to, we divided the cytoplasmic tail into subdomains (*Figure 2B*) and, initially, created a series of sequential truncations within the iRhom2 cytoplasmic tail (*Figure 2C*). Then, we created more focussed deletions within the area identified in the previous experiment (*Figure 2D*). This revealed that amino acids 191–271 of the iRhom2 tail contain the main determinant for iTAP binding (*Figure 2C,D*). Further studies are required to assess whether the specific binding of iTAP to iRhom is direct, or via an intermediary.

We next examined the cellular localization of GFP-tagged iTAP in fixed and permeabilized mammalian cells. As shown in *Figure 2—figure supplement 1A*, iTAP-GFP exhibited a powdery staining in the cytoplasm and nucleus. When co-expressed with mCherry-tagged iRhom2, iTAP was recruited to areas of iRhom2 staining (*Figure 2—figure supplement 1B*).

## iTAP-deficient cells are impaired in the shedding of TACE substrates

To determine the functional importance of iTAP binding to iRhom, we used CRISPR to ablate iTAP in HEK 293ET cells, which was confirmed by the lack of iTAP protein expression (*Figure 3A*). As

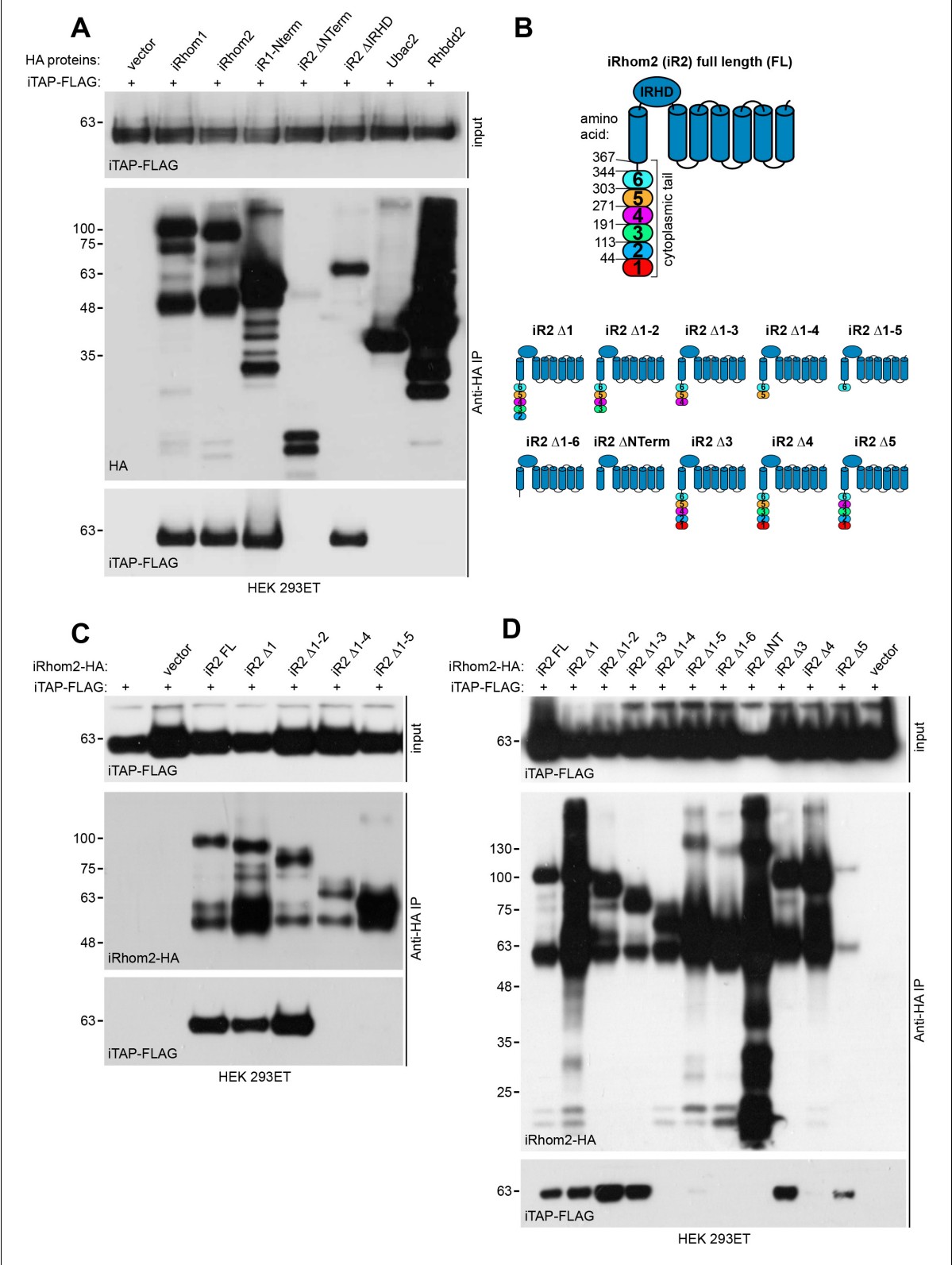

**Figure 2.** Validation of iTAP binding to iRhoms. (**A**). iTAP binds specifically to iRhom −1 and −2. Human iTAP-FLAG was transfected into HEK 293ET cells alongside empty vector or the indicated HA-tagged iRhoms, their deletion mutants or the rhomboid pseudoproteases Ubac2 or Rhbdd2. HA proteins were immunoprecipitated and FLAG binding was assessed by western blot. (**B**). Schematic diagram indicating truncation mutants of the iRhom2 cytoplasmic tail that were generated to map the iTAP binding region. The cytoplasmic tail of iRhom2 was divided into six arbitrary portions. (**C**,
*Figure 2 continued on next page*

*Figure 2 continued*

D). iTAP binds to iRhom2 within an area defined by region 4 of iRhom2 (aa 192-271). HEK 293ET cells were transfected with iTAP-FLAG and iRhom2-HA full length (FL) or the indicated iRhom2-HA deletion constructs shown in (B). Anti-HA immunoprecipitates were assessed for the binding of iTAP-FLAG by western blotting.

DOI: https://doi.org/10.7554/eLife.35032.005

The following figure supplement is available for figure 2:

**Figure supplement 1.** Cellular localization of iTAP.

DOI: https://doi.org/10.7554/eLife.35032.006

TACE trafficking and cell surface stimulation depends on iRhoms, we examined the ability of WT versus iTAP-null cells to support release of TACE substrates. Notably, the PMA-induced shedding of a panel of chimeric alkaline phosphatase (AP) TACE substrates (*Sahin et al., 2004*), including EGFR ligands and TNF, was substantially impaired in iTAP KO cells (*Figure 3B*). This shedding defect was rescued by the expression of an iTAP cDNA, confirming that the loss of iTAP was directly responsible for defective TACE activity (*Figure 3C,D*). To test the hypothesis that the basis for the shedding defects was in fact reduced TACE proteolytic activity in iTAP KO cells, we assayed TACE enzymatic activity directly using a peptide substrate (*Figure 3—figure supplement 1*). As expected, TACE immunoprecipitates (*Figure 3—figure supplement 1A*) from iTAP KO cells exhibited substantially depleted levels of TACE activity (*Figure 3—figure supplement 1B*), confirming that the loss of iTAP specifically impairs TACE rather than affecting its substrates.

Our previous studies have shown that iRhom proteins are highly specific regulators of TACE that do not affect the trafficking/activity of related proteases in the ADAM metalloprotease family, including ADAM10, the closest relative of TACE (*Adrain et al., 2012*; *Christova et al., 2013*). To examine whether iTAP was similarly dedicated specifically to the iRhom/TACE pathway we examined whether cleavage of the EGFR ligands EGF and BTC, which are cleaved specifically by ADAM10 (*Sahin et al., 2004*), was affected by loss of iTAP. Notably the cleavage of these ADAM10 substrates was unaffected (*Figure 3E*) while the release of a model secreted substrate was similarly unimpaired in iTAP KO cells (*Figure 3F*). These data confirm that iTAP is a highly specific regulator of the TACE pathway that does not affect secretion in general.

## Mature TACE is specifically reduced in iTAP-deficient cells

To investigate how loss of iTAP affected TACE so profoundly, we examined the maturation status of TACE, a readout for its trafficking and activation status (*Adrain et al., 2012*) in iTAP KO cell lines (*Figure 4A*). As a positive control, we included lysates from iRhom1/iRhom2 double knockout MEFs which completely lack mature TACE (*Christova et al., 2013*). Although a few experiments showed an overall reduction in the TACE levels in iTAP KO cells, the consistent and most pronounced phenotype, found in all iTAP-null cell lines, was a dramatic depletion of mature TACE, identified by its faster migration pattern (*Figure 4B*). As TACE is heavily glycosylated, to confirm this observation more clearly, we treated lysates with the deglycosylating enzymes Endoglycosidase-H (Endo-H),

**Table 1.** iTAP interacts with endogenous iRhoms.

Lysates from HEK 293ET cells expressing iTAP-FLAG versus cells containing empty vector or expressing a panel of control proteins (TNF-FLAG, STING-FLAG, SREBP2-FLAG) were immunoprecipitated for FLAG and subjected to mass spectrometry. Peptides assigned to iRhom1 or iRhom2 are found specifically in iTAP precipitates.

| iRhom1 | Vector | iTAP | TNF | STING | SREBP2 |
|---|---|---|---|---|---|
| Peptide counts | 0 | **27** | 0 | 0 | 0 |
| Sequence coverage [%] | 0 | **37.7** | 0 | 0 | 0 |
| iRhom2 | Vector | iTAP | TNF | STING | SREBP2 |
| Peptide counts | 3 | **44** | 2 | 2 | 2 |
| Sequence coverage [%] | 2.9 | **45.4** | 1.8 | 2.2 | 2 |

DOI: https://doi.org/10.7554/eLife.35032.007

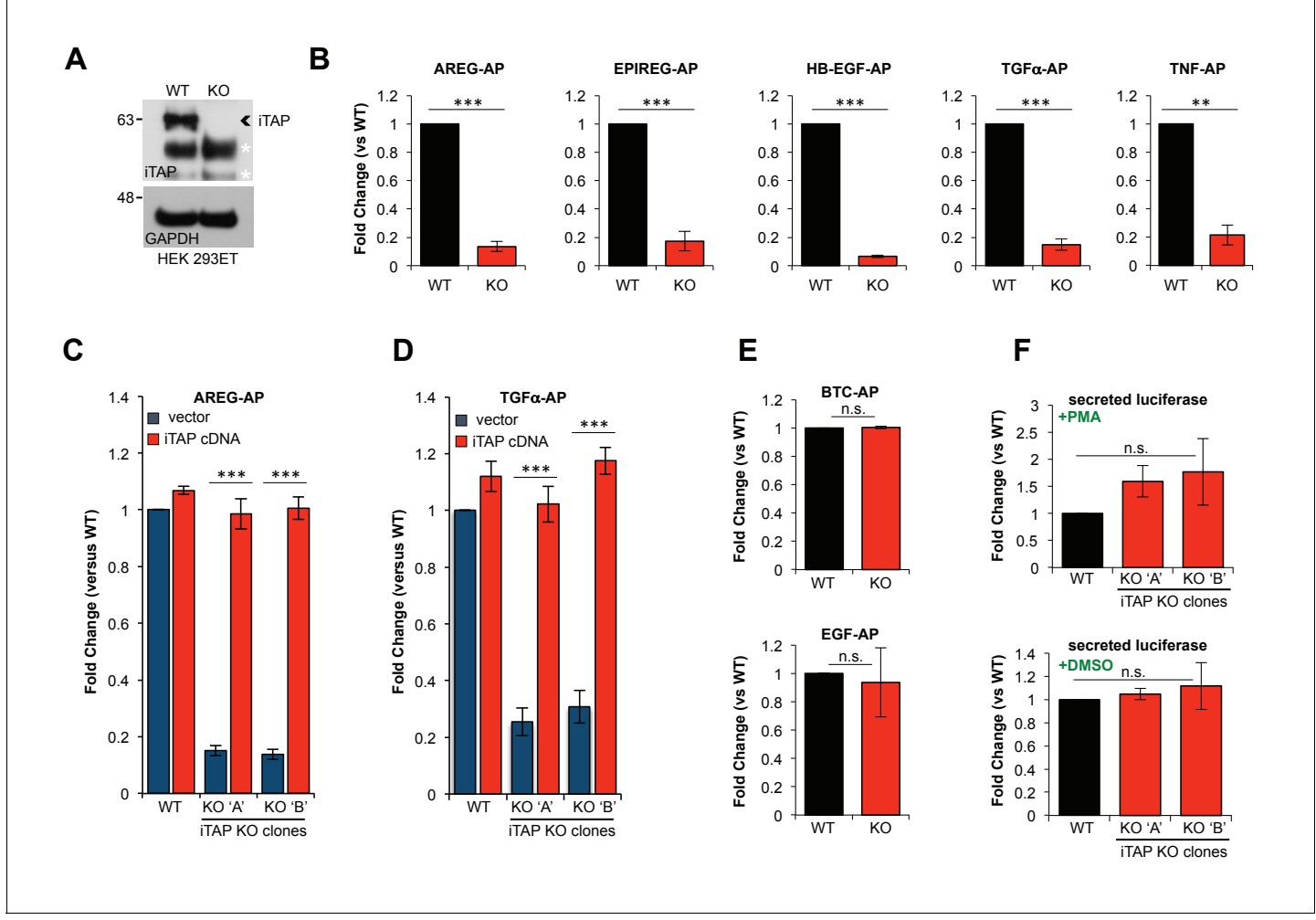

**Figure 3.** KO of iTAP diminishes TACE proteolytic activity. (**A**). Anti-iTAP immunoprecipitates from WT versus iTAP KO HEK 293ET cells were analyzed by immunoblotting. A GAPDH blot is the loading control for the inputs. Non-specific bands are indicated by white asterisks. (**B**). PMA-stimulated TACE shedding is impaired in iTAP KO cells. The TACE substrates [Amphiregulin (AREG), Epiregulin (EPIREG), Heparin Binding-Epidermal Growth Factor (HB-EGF), Transforming Growth Factor-α (TGFα) and Tumor Necrosis Factor (TNF)] fused to alkaline phosphatase (AP) were transfected into HEK 293ET WT or iTAP KO cells. TACE activity was assessed based on AP activity secreted into the supernatant of the cells as described in materials and methods. (**C, D**). Expression of iTAP rescues the impaired shedding in iTAP KO cells. WT or iTAP KO HEK 293ET cells stably expressing empty vector or human iTAP were transfected with AREG-AP or TGFα-AP, then challenged in PMA shedding assays as described above. (**E**). The shedding impairment is specific to TACE. ADAM10 AP-fused substrates [Betacellulin (BTC) and Epidermal Growth Factor (EGF)] were transfected into the WT vs iTAP KO HEK 293ET cells. The cells were treated with the ADAM10 stimulant Ionomycin (IO) and AP activity was measured in the medium. (**F**). Global secretion is not impaired in iTAP KO cells. WT or iTAP KO cells were transfected with secreted luciferase and luciferase-associated luminescence was measured in the supernatant of PMA-stimulated cells (upper graph) or vehicle (DMSO, lower graph). Here and throughout: KO 'A' and KO 'B' denotes independent iTAP KO HEK 293ET clones. PMA (1 μM) or IO (2.5 μM) incubations took place for 1 hr following serum starvation. Shedding or secretion values are expressed as fold change relative to WT cells. Data are presented as mean ± standard deviation and represent at least three independent experiments. *=p ≤ 0.05, **=p ≤ 0.01, ***=p ≤ 0.001 and n.s. = non significant.

DOI: https://doi.org/10.7554/eLife.35032.008

The following source data and figure supplements are available for figure 3:

**Source data 1.** PMA-stimulated TACE shedding is impaired in iTAP KO cells.
DOI: https://doi.org/10.7554/eLife.35032.011

**Source data 2.** Expression of iTAP rescues the impaired shedding in iTAP KO cells.
DOI: https://doi.org/10.7554/eLife.35032.012

**Source data 3.** The shedding impairment is specific to TACE.
DOI: https://doi.org/10.7554/eLife.35032.013

**Source data 4.** Global secretion is not impaired in iTAP KO cells.
DOI: https://doi.org/10.7554/eLife.35032.014

*Figure 3 continued on next page*

*Figure 3 continued*

**Figure supplement 1.** Assessment of proteolytic activity in TACE IPs from WT versus iTAP KO cells.

DOI: https://doi.org/10.7554/eLife.35032.009

**Figure supplement 1—source data 1.** iTAP ablation impairs the performance of TACE in activity assays.

DOI: https://doi.org/10.7554/eLife.35032.010

which removes high mannose N-linked glycans added in the ER, but not complex N-linked glycans found in the later secretory pathway, versus PNGase F, which deglycosylates both (*Figure 4C,D*). This confirmed that iTAP KO cell lines were substantially depleted of mature TACE (*Figure 4B,D,E*), which could be rescued specifically by iTAP overexpression in iTAP KO cells (*Figure 4E*). Overexpression of iTAP in WT cells also modestly enhanced mature TACE levels (*Figure 4E*) and densitometric analysis confirmed once again that the loss, or reintroduction, of iTAP most profoundly affected mature TACE levels (*Figure 4E*, graphs).

A clear prediction from these experiments is that iTAP-null cells should lack mature, cell surface TACE, explaining the basis of the proteolytic defects observed (*Figure 3*, *Figure 3—figure supplement 1A,B*). We tested this hypothesis in experiments with a non-cell permeable biotinylated crosslinker, which revealed drastically reduced mature TACE levels on the cell surface (*Figure 4F*). Finally, as predicted by stringent specificity of iTAP for TACE, the maturation of other related ADAM proteases was unimpaired (*Figure 4G*). Together these data confirm that iTAP is a dedicated regulator of the iRhom/TACE sheddase complex.

## iTAP is required to maintain iRhom2 stability in the late secretory pathway

The observation that iTAP-null cells exhibited drastically depleted mature TACE levels could be explained by two potential mechanisms. First, loss of iTAP, which binds to iRhom2 (*Figures 1–2*; *Table 1*), could impair ER exit of the iRhom/TACE complex, causing a failure in TACE maturation, as observed in iRhom KO cells (*Adrain et al., 2012*; *Christova et al., 2013*). Alternatively, as TACE undergoes constitutive recycling from the plasma membrane (*Dombernowsky et al., 2015*) and iRhom2 traffics to the cell surface and enters the endolysosomal pathway (*Grieve et al., 2017*; *Cavadas et al., 2017*; *Maney et al., 2015*), iTAP could stabilize iRhom/TACE complexes on the plasma membrane or within the endocytic pathway. Given the established role of FERM domain proteins in stabilizing proteins on the cell cortex, this second possibility seemed plausible.

To investigate the impact of iTAP on iRhom2 stability, we first used RAW 264.7 cells. These macrophage-like cells express high levels of endogenous iRhom2, making its detection more feasible than in HEK 293ET cells or MEFs. Strikingly, endogenous iRhom2 was depleted in iTAP KO RAW 264.7 cells (*Figure 5A*), indicating that iTAP is essential to maintain iRhom2 stability. Consistent with this, in HEK 293ET cells, iTAP transient overexpression increased steady state levels of overexpressed iRhom2-HA and enhanced the half-life (see graph, *Figure 5B*) of the protein, during a timecourse of cycloheximide (CHX) treatment, used to block additional protein synthesis (*Figure 5B*). This experiment, in which iRhom2 was expressed from an artificial promoter, indicates that the impact of iTAP on iRhom2 levels is independent of transcription. As anticipated by these results, transiently overexpressed iRhom2-HA was also destabilized in iTAP KO cells (*Figure 5—figure supplement 1A*). Consistent with the ability of iTAP to impact profoundly on iRhom2 stability, we observed striking colocalization between mCherry-iRhom2 and iTAP-GFP, when they were coexpressed in HeLa cells, as judged by Pearson's correlation and Manders' overlap coefficients (*Figure 5C*).

These colocalization data indicate that iRhom and iTAP interact in multiple compartments (*Figure 5C*), including the ER and plasma membrane. Two major possibilities exist to explain the decreased half-life of iRhom2 in iTAP KO cells: in the absence of iTAP, iRhom2 may be degraded in the early secretory pathway (the ER), or in the late secretory pathway (lysosomes). To address this, we used Endo-H deglycosylation to discriminate between the impact upon iTAP overexpression on the Endo-H-sensitive pool of iRhom2 in the ER, versus the Endo-H-insensitive fraction that has entered the later secretory pathway. As anticipated (*Zettl et al., 2011*), most overexpressed iRhom2 is Endo-H sensitive, hence still located within the early secretory pathway (*Figure 5D*). Strikingly, the

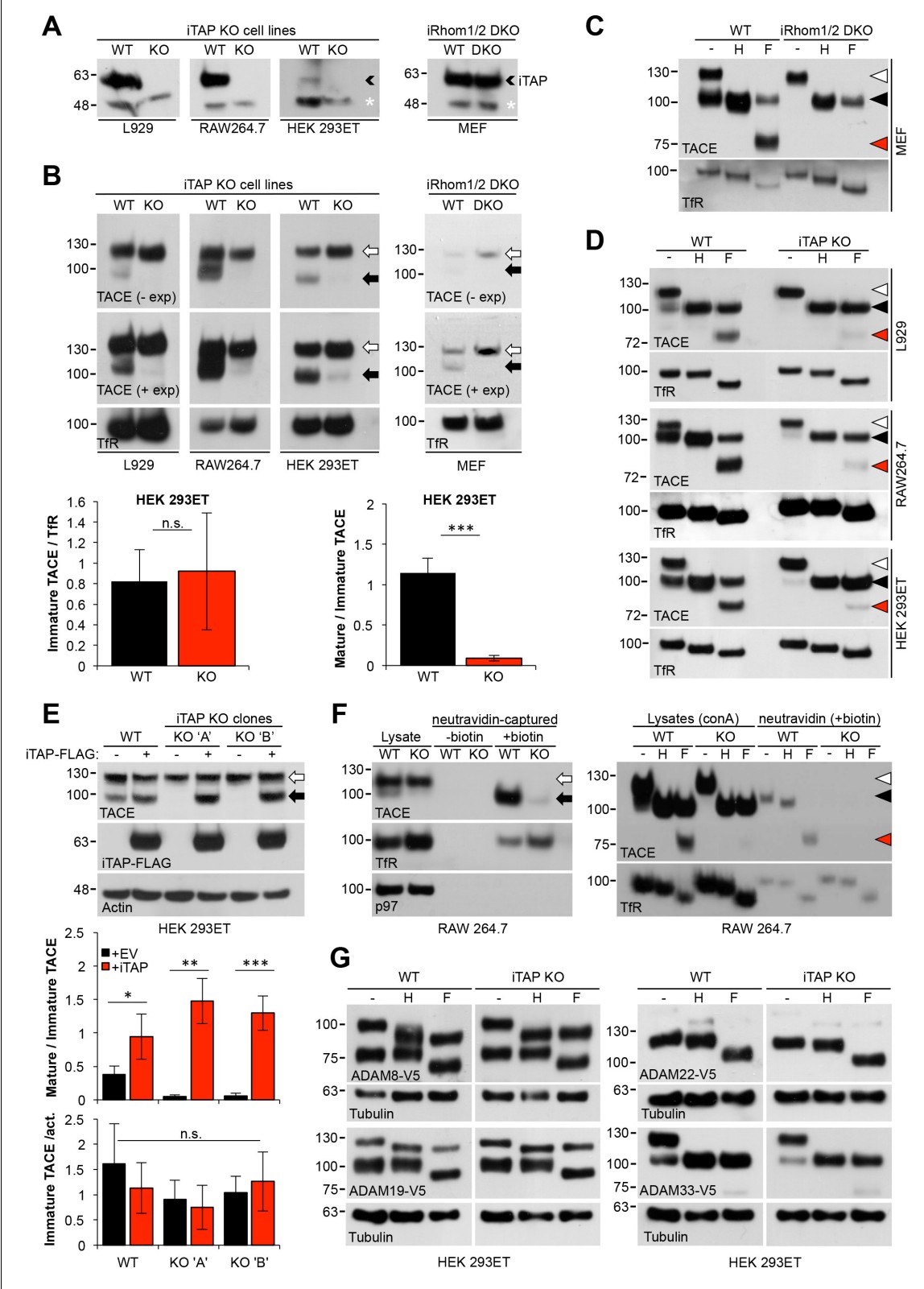

**Figure 4.** Mature TACE is specifically depleted in iTAP KO cells. (**A**). iTAP was knocked out in L929, RAW 264.7 and HEK 293ET cells using CRISPR. Lysates were immunoblotted with anti-iTAP antibodies. A small black arrowhead indicates iTAP protein whereas a non-specific band (white asterisk) serves as a loading control. (**B**). Glycoproteins from lysates isolated from the cells in (**A**) were enriched using concanavalin A-sepharose (conA) and TACE levels were assessed by western blot. Here and throughout, the immature form of TACE is indicated by a white arrow, whereas, the mature form

*Figure 4 continued on next page*

Figure 4 continued

is denoted by a black arrow. iRhom double KO MEFs were used as a reference and the transferrin receptor (TfR) as a loading control. Lower panels: densitometry in HEK 293ET. Left hand panel: Levels of immature TACE normalized to TfR. Right hand panel: levels of mature TACE as a relative proportion of immature TACE in WT and iTAP KO HEK 293ET. (C,D). Validation of mature and immature TACE detection in panels of WT versus iRhom2 DKO (C) or iTAP KO (D) cells, by deglycosylation. ConA enriched lysates from the cell lines in (A) were treated with endoglycosidase H (Endo-H; H; which cleaves ER-resident glycans only) and PNGase F (F; which cleaves both ER and post-ER glycans). Here and throughout: the immature TACE is indicated with white arrowheads; the black arrowhead denotes both glycosylated mature TACE and deglycosylated immature TACE respectively (which have similar electrophoretic mobility), whereas red arrowheads denote the fully deglycosylated, mature, TACE polypeptide. (E). iTAP expression restores the presence of mature TACE in iTAP KO cells. Lysates from WT or iTAP KO HEK 293ET stably expressing empty vector (-, EV) or human iTAP (+) were screened for mature TACE. Actin was used as a loading control. Middle and lower panels: densitometric analysis indicates that iTAP expression increases the levels of mature TACE but does not affect the levels of immature TACE. Middle panel: levels of mature TACE as a relative proportion of immature TACE in WT and KO upon iTAP or EV expression in WT and iTAP KO HEK 293ET clones. Lower panel: Levels of immature TACE after normalization to actin. (F). iTAP KO cells lack mature cell surface TACE. Left hand panel: RAW 264.7 WT or iTAP KO were surface-biotinylated *in vivo* and lysates were enriched for biotinylated proteins with neutravidin resin. Probing for TfR was used as a cell surface positive control protein whereas anti-p97 probing demonstrates that intracellular proteins were not labeled. Right hand panel: Cell surface biotinylated proteins were deglycosylated using Endo-H (H) or PNGase F (F). ConA enriched lysates were run as mobility controls, for immature and mature TACE. Blots were probed for TACE and for TfR as a control protein. (G). Loss of iTAP has no impact on the mature species of other ADAM metalloproteases. HEK 293ET WT or KO cells were transfected with the indicated panel of V5-tagged ADAMs. The lysates were deglycosylated as described above and Tubulin serves as a loading control. Throughout: Data are presented as mean ± standard deviation and represent three independent experiments. *=p ≤ 0.05, **=p ≤ 0.01, ***=p ≤ 0.001 and n.s. = non significant.

DOI: https://doi.org/10.7554/eLife.35032.015

The following source data is available for figure 4:

**Source data 1.** iTAP KO cells are depleted in mature TACE levels.
DOI: https://doi.org/10.7554/eLife.35032.016

**Source data 2.** iTAP expression restores the presence of mature TACE in iTAP KO cells.
DOI: https://doi.org/10.7554/eLife.35032.017

co-expression of iTAP increased the overall levels of iRhom2, but selectively enriched the post-ER fraction of iRhom2 (*Figure 5D*), suggesting a disproportionate impact on the form of iRhom2 that had traversed to the late secretory pathway.

To obtain additional insights, we used a binding assay, coupled to deglycosylation analysis, to compare which species of iRhom2 bind to iTAP. Cells were treated with or without DSP, a thiol-reducible cell-permeable cross-linker, to covalently trap complexes *in situ* (*Adrain et al., 2012*), enabling us to discriminate between interaction *in vivo*, compared to potential adventitious binding post-lysis. After immunoprecipitation, samples were treated with DTT to reverse the cross-linking. Notably, compared to IPs done without cross-linking, iTAP IPs from cross-linked cells showed a clear enrichment for post-ER (Endo-H insensitive) iRhom2 (*Figure 5E*), although the ER-localized form of iRhom2 was also readily detected. Taken together with the colocalization data (*Figure 5C*), we propose that the loading of iTAP onto the sheddase complex occurs already in the ER but the binding is sustained throughout the sheddase complex's itinerary in the late secretory pathway, where iTAP's affinity for iRhom2 appears to be higher. These data are consistent with the finding that iTAP selectively affects mature TACE (*Figure 4*). Finally, ruling out a requirement for iTAP in controlling the ER-to-Golgi trafficking of iRhom2, we found that the ER exit of iRhom2 was unimpaired in iTAP KO cells, judged by the presence of Endo-H-resistant iRhom2 (*Figure 5F*; *Figure 5—figure supplement 1B*). Hence, although iTAP binds to the sheddase complex in the early secretory pathway, its impact appears to be more decisive in the late secretory pathway.

## iTAP modulates the plasma membrane stability of iRhom2 and TACE by preventing their aberrant sorting to the lysosome

As ablation of iTAP results in dramatically reduced iRhom2 levels (*Figure 5A*) and iTAP increases the abundance of post-ER iRhom2 (*Figure 5D,E*), we hypothesized that iTAP controls the stability of iRhom2 in the late secretory pathway. Consistent with this premise, cell surface biotinylation experiments revealed that iTAP expression increased the steady state levels of cell surface iRhom2, and prolonged its cell surface stability, when CHX was used to block additional protein synthesis (*Figure 5G*). In further agreement, the co-expression of iTAP with GFP-iRhom2 in MEFs enhanced the amount of GFP-iRhom2 detected on the plasma membrane (*Figure 5H*), supporting the premise

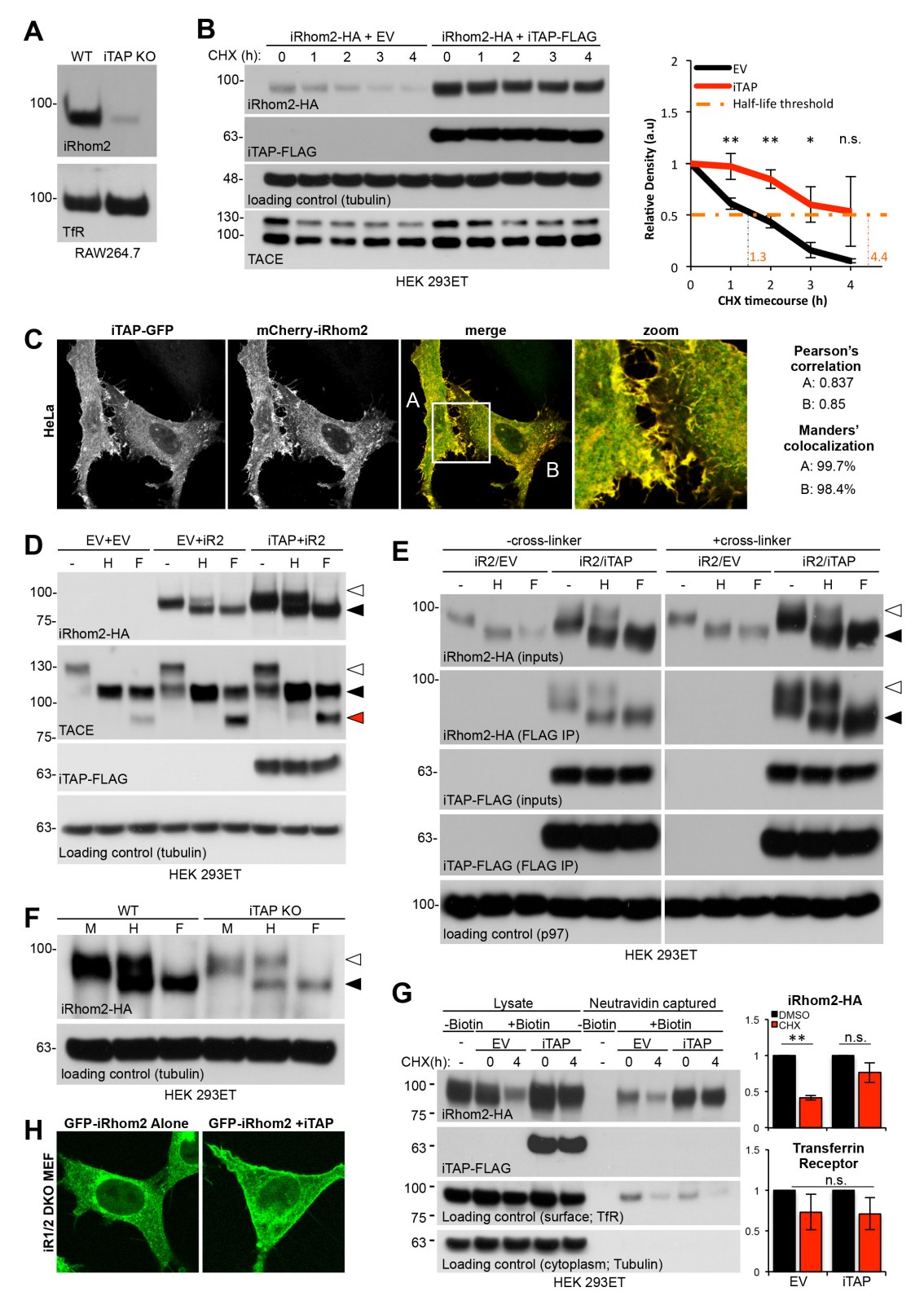

**Figure 5.** iTAP is required to promote iRhom stability at the cell surface. (**A**). iRhom2 is depleted in iTAP KO cells. Lysates from WT vs iTAP KO RAW 264.7 were probed for endogenous iRhom2. The transferrin receptor (TfR) is a loading control. (**B**). iTAP expression enhances the stability of iRhom2. Stable iRhom2-HA-expressing HEK 293ET were transiently transfected with empty vector (EV) or iTAP-FLAG. 48 hr post-transfection, the cells were treated with 100 µg/mL Cycloheximide (CHX) for the indicated durations. The stability of iRhom2 was assessed by HA blotting. The graph to the right

*Figure 5 continued on next page*

Figure 5 continued
of the panel indicates the relative density of iRhom2-HA bands from cells expressing EV (black) versus iTAP (red). The half-life of iRhom2 under both conditions is calculated. (C). iTAP and iRhom2 co-localize. HeLa cells were transfected with iTAP-GFP and mCherry-iRhom2. Two areas, A and B, were selected for the calculation of the Pearson's correlation and Manders' colocalization co-efficients, respectively. (D). iTAP expression enhances the post-ER form of iRhom2. HEK 293ET expressing stably iRhom2-HA minus or plus stably expressed iTAP-FLAG were deglycosylated with Endo-H or PNGase F. (E). iRhom2-HA stably expressing HEK 293ET cells were transiently transfected with EV or iTAP-FLAG. Cells were treated ± the thiol-reducible cell-permeable crosslinker, DSP, and then anti-FLAG immunoprecipitations were performed from the lysates. Prior to SDS-PAGE and immunoblotting, lysates and co-immunoprecipitates were denatured in the presence of DTT to break the DSP-mediated covalent cross-links. Samples containing iRhom2-HA were deglycosylated as described before. (F). ER exit of iRhom2 is not impaired in iTAP KO cells. WT or iTAP KO HEK 293ET were transiently transfected with iRhom2-HA. Their lysates were deglycosylated as described. Endo-H–sensitive (black arrowhead) and –insensitive (white arrowhead) bands are noted. (G). iTAP expression stabilizes iRhom2 on the cell surface. The same cell lines as in (B), (E), were subject to a cell surface biotinylation protocol and the cell surface levels of iRhom2 in response to CHX treatment were evaluated. The graphs on the right hand side show densitometric analysis of the surface fractions of iRhom2-HA (upper graph) or TfR (lower graph) (H). iRhom1/2 DKO MEFs stably expressing mouse eGFP-iRhom2 either alone or together with mouse iTAP-mCherry were imaged as live cells. The eGFP-iRhom2 signal is shown.

DOI: https://doi.org/10.7554/eLife.35032.018

The following source data and figure supplement are available for figure 5:

**Source data 1.** iTAP expression enhances the stability and half-life of iRhom2.
DOI: https://doi.org/10.7554/eLife.35032.020
**Source data 2.** iTAP expression stabilizes iRhom2 on the cell surface.
DOI: https://doi.org/10.7554/eLife.35032.021
**Figure supplement 1.** iTAP ablation increases iRhom2 degradation but doesn't affect its ER exit.
DOI: https://doi.org/10.7554/eLife.35032.019

that iTAP promotes the cell surface stability of iRhom2. Together, our data indicate that iTAP's primary function is to stabilize the sheddase complex on the cell surface, reconciling the pronounced loss of mature TACE in iTAP KO cells and the increased binding propensity of iTAP for post-ER iRhom2.

We next addressed the functional basis for the pronounced loss of iRhom2 and mature TACE in iTAP KO cells. To test the hypothesis that loss of iTAP triggers the degradation of iRhom2 and TACE in lysosomes, the major degradative compartment in the late secretory pathway, we examined the localization of mCherry-iRhom2 in WT or iTAP KO HeLa cells derived by CRISPR (*Figure 6—figure supplement 1*). As shown in *Figure 6A*, in WT HeLa cells, mCherry-iRhom2 did not co-stain appreciably with lysosomes. By sharp contrast, iTAP ablation resulted in a pronounced co-localization of iRhom2 with the lysosomal marker LAMP2 (Lysosomal-Associated Membrane Protein 2), indicating mis-sorting of iRhom2 to lysosomes. This phenotype was specific since the co-transfection of iTAP-GFP into iTAP KO HeLas rescued the aberrant accumulation of mCherry-iRhom2 in lysosomes (*Figure 6B*), resulting in the marked co-localization of iRhom and iTAP observed previously (*Figure 2—figure supplement 1B*; *Figure 5C*). Consistent with these observations, a panel of lysosomotropic drugs that inhibit lysosomal proteolysis by impairing lysosomal acidification, rescued iRhom2 stability in iTAP KO cells (*Figure 6C*).

The rescue of mature TACE under similar conditions was more modest (*Figure 6D* and data not shown), perhaps because of the slow trafficking time of TACE in the secretory pathway (*Schlöndorff et al., 2000*) and because iTAP acts directly on iRhoms. Consistent with the notion that iRhom2 can influence the routing of TACE into lysosomes, we found that overexpressed TACE-GFP was only recruited into lysosomes when sufficient iRhom2 was co-overexpressed (*Figure 6E*). In conclusion, our data reveal that when the normal stoichiometric ratio of iTAP to iRhom2 is disrupted (e.g. upon iRhom2 overexpression or iTAP ablation), iRhom2 is mis-sorted into the lysosome, then degraded. This highlights an important physiological role for iTAP in maintaining the cell surface stability of the iRhom2/TACE sheddase complex.

## iTAP controls the stability of mature TACE, in mice

iTAP is expressed in a range of mouse tissues relevant to iRhom and TACE biology (*Peschon et al., 1998*; *Li et al., 2015*) (*Figure 1—figure supplement 1*). As the experiments conducted so far focused on transformed cell lines, we next examined the physiological importance of iTAP at the organismal level. To examine the role of iTAP in mice, we generated a mutant in which the first

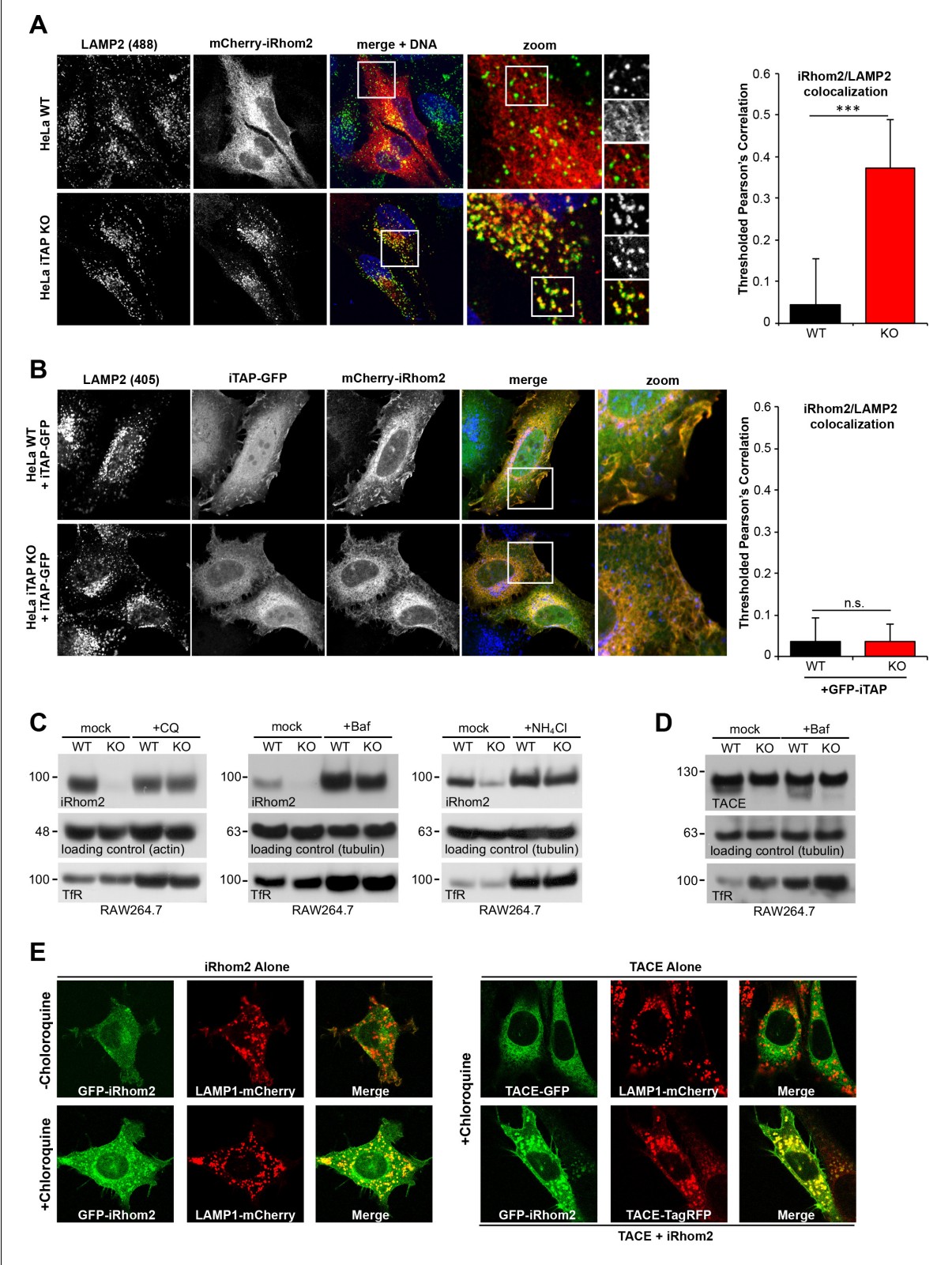

**Figure 6.** iTAP is required to prevent the trafficking of the sheddase complex to the lysosomes, where iRhom2/TACE are degraded. (**A**). WT or iTAP KO HeLa cells were transfected with mCherry-iRhom2. Fixed cells were immunostained for the lysosomal marker LAMP2 and stained with DAPI. iRhom2/LAMP2 co-localization was quantified (right-hand graph). (**B**). The phenotype observed in (**A**) is reverted upon the co-expression of iTAP-GFP, resulting in no co-localization of mCherry-iRhom2 with endogenous LAMP2. (**C**). WT or iTAP KO RAW 264.7 cells were treated with 50 μM Chloroquine (CQ) for

*Figure 6 continued*

48 hr, 100 μM Bafilomycin (Baf) for 16 hr, or 10 mM ammonium chloride (NH₄Cl) for 48 hr and endogenous iRhom2 levels were detected by western blotting. Actin or tubulin were used as loading controls and the Transferrin Receptor (TfR) acts as a control for the inhibition of lysosomal hydrolases. (D). Lysates from cells treated with Baf as described above, were conA-enriched and probed for TACE on a western blot. Tubulin and TfR are controls for loading and lysosomal inhibition, respectively. (E). In the left-hand panel, eGFP-iRhom2 was stably expressed in WT MEFs (expressing endogenous iTAP) in the presence of lysosomal marker LAMP1-mCherry, and the subcellular localization of both proteins was imaged in live cells using confocal microscopy in the absence (upper row) or presence (lower row) of 10 μM Chloroquine. The results indicate that iRhom2 alone is trafficked into the lysosomes. In the right-hand panel, WT MEFs stably expressing TACE-GFP and LAMP1-mCherry in the presence of 10 μM Chloroquine were imaged similarly. The results indicate that TACE alone does not localize in lysosomes (upper row). However, lysosomal trafficking of TACE-TagRFP is induced by the presence of co-expressed eGFP-miRhom2 (lower row).
DOI: https://doi.org/10.7554/eLife.35032.022

The following source data and figure supplement are available for figure 6:

**Source data 1.** Quantification of mCherry-iRhom2/LAMP2 colocalization analyses.
DOI: https://doi.org/10.7554/eLife.35032.024
**Figure supplement 1.** Validation of iTAP KO in HeLa cells.
DOI: https://doi.org/10.7554/eLife.35032.023

coding exon (exon 2) of the *Frmd8* (iTAP) gene was deleted by CRISPR (*Figure 7A*, *Figure 7—figure supplement 1*). MEFs isolated from iTAP KO embryos lacked iTAP protein expression, confirming the successful targeting of the *Frmd8* gene (*Figure 7B*). As anticipated, MEFs from two independent iTAP KO embryos exhibited the characteristic pronounced depletion of mature TACE levels (*Figure 7C*).

Focussing next on potential phenotypes of the iTAP–null mouse mutants themselves, we harvested tissues from iTAP KO mice to assess the maturation status of TACE (*Figure 7D*). Significantly, with the possible exception of skin, where iTAP loss may be mitigated by other molecules, we observed a substantial depletion in the relative proportion of mature TACE in a range of iTAP KO mouse tissues, and in primary macrophages isolated from the bone marrow of iTAP KOs. These data reinforce the notion that iTAP is an important physiological regulator of the iRhom/TACE/TNF axis *in vivo*, making it important to dissect fully, in future, the organismal role of iTAP.

## iTAP is a physiological regulator of TNF release in humans

As the PMA-stimulated release of a chimeric alkaline phosphatase-tagged TNF was impaired in iTAP KO cells (*Figure 3B*), we hypothesized that iTAP was an important physiological regulator of TNF secretion. To test this, we isolated primary monocytes from peripheral human blood, then induced the differentiation of these cells to primary human macrophages. Notably, the stimulated release of endogenous TNF in response to lipopolysaccharide in these cells was profoundly impaired, when iTAP expression was ablated by specific siRNAs (*Figure 7E*). As expected, secretion of IL-6 and IL-8, which is TACE-independent, was unaffected (*Figure 7E*). Our data indicate that iTAP is an essential physiological regulator of TNF secretion in primary human macrophages, the principal source of secreted TNF in vivo.

## Discussion

Our work identifies iTAP as an important physiological regulator of the iRhom2/TACE sheddase complex, which is essential for the secretion of TNF and for a panoply of other substrates including ligands of the epidermal growth factor receptor. Our current and previous data (*Cavadas et al., 2017*) suggest that trafficking to, and degradation within, the lysosome is a default itinerary incurred by iRhom2, and that iRhom2 potentially encodes the determinants that lead to the default trafficking of iRhom2 and TACE to the lysosome. We now show that iTAP is essential to stabilize iRhom2 on the cell surface, preventing the routing of the sheddase complex to the lysosome, and licensing TACE to cleave its substrates for signaling (summarized in *Figure 8*). iTAP, hence, emerges as an important regulator of inflammation and growth factor signaling, during development, normal physiology, infection and inflammatory disease.

An obvious question concerns the extent to which the established features and roles of FERM domain proteins apply to iTAP and hence to the regulation of the iRhom/TACE pathway. A general

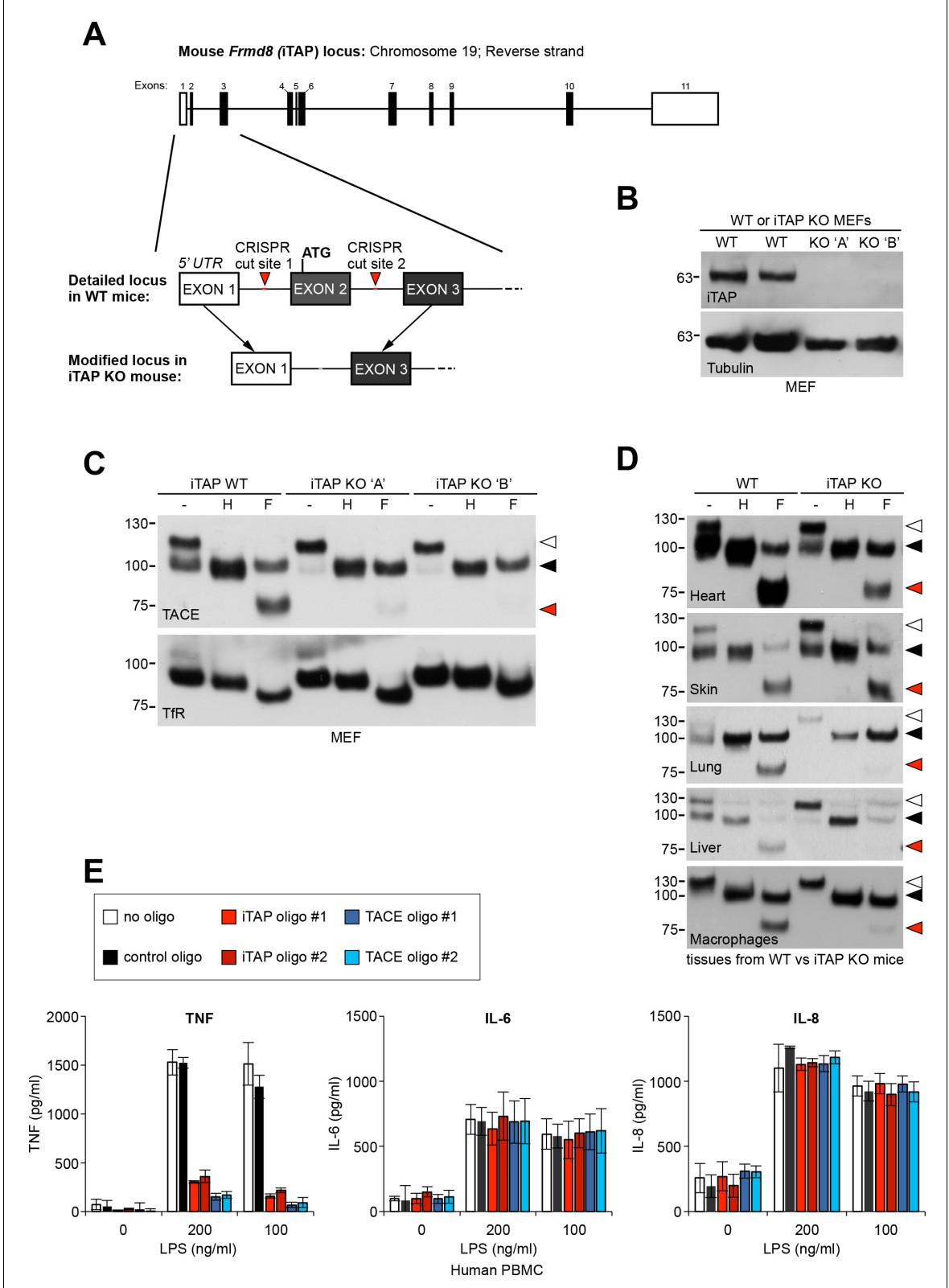

**Figure 7.** iTAP is essential for TACE maturation and function in primary cells and tissues from human and mouse. (**A**). Schematic representation of the CRISPR targeting strategy to delete mouse *Frmd8* (iTAP) gene using two guide RNAs flanking the first coding exon (exon 2). In the upper schematic of the *Frmd8* locus, open boxes indicate non-coding exons whereas filled boxes indicate coding exons (**B**). Mouse embryonic fibroblasts (MEFs) were isolated from WT versus two independent iTAP KO E14.5 embryo littermates. The loss of iTAP at the protein level is shown by immunoblotting. (**C**).
*Figure 7 continued on next page*

*Figure 7 continued*

Mature TACE is diminished in iTAP KO MEFs. ConA-enriched lysates from MEFs isolated from WT versus iTAP KO embryos were deglycosylated as described previously. The transferrin receptor (TfR) is used as a loading control. (D). Mature TACE is depleted or diminished in TACE-relevant tissues from iTAP KO mice. ConA-enriched lysates from WT vs iTAP KO mouse tissues and bone marrow-derived macrophages, were deglycosylated as described previously. TACE was detected by western blot. The immature and mature species of TACE are indicated with white arrowheads and black arrowheads respectively, whereas red arrowheads denote the fully deglycosylated mature polypeptide. The experiment was performed twice with lysates isolated from tissues from two individual KO mice. (E). iTAP is essential for TACE physiological regulation in human primary cells. Isolated primary human peripheral blood mononuclear cells (PBMC) were differentiated into monocytes, then electroporated with the indicated siRNAs. Cells were then stimulated with the indicated concentrations of lipopolysaccharide (LPS). After 18 hr, the concentration of the cytokines TNF, IL-6 and IL-8 secreted into the supernatants, was measured by ELISA. The experiment was done three independent times and data from one representative experiment is shown. Data presented as mean ± standard error from triplicate measurements.

DOI: https://doi.org/10.7554/eLife.35032.025

The following source data and figure supplement are available for figure 7:

**Source data 1.** iTAP is essential for TNF secretion in primary macrophages.
DOI: https://doi.org/10.7554/eLife.35032.027
**Source data 2.** iTAP is not essential for IL-6 secretion.
DOI: https://doi.org/10.7554/eLife.35032.028
**Source data 3.** iTAP is not essential for IL-8 secretion.
DOI: https://doi.org/10.7554/eLife.35032.029
**Figure supplement 1.** Mouse *Frmd8*/iTAP gene targeting via CRISPR.
DOI: https://doi.org/10.7554/eLife.35032.026

theme is that FERM-domain proteins connect the cytoplasmic tails of cell surface client proteins to the cortical actin cytoskeleton to enhance their stability (*Hoover and Bryant, 2000*; *Baines et al., 2014*; *Moleirinho et al., 2013*). While iTAP binds to the cytoplasmic tails of iRhoms, which are found on the plasma membrane, our preliminary experiments failed to detect robust binding of iTAP to actin (*Figure 8—figure supplement 1A*). Besides, we have not identified predicted actin binding motifs in the C-terminus of iTAP. Future experiments will be required to determine precisely how iTAP stabilizes iRhom and TACE in the late secretory pathway.

Some FERM-domain proteins are implicated in endosomal sorting, the process whereby endocytosed proteins are sorted in early endosomes, for routing to the multi-vesicular body, lysosome, *trans*-Golgi network, recycling endosome or, alternatively, 'fast' recycling back to the cell surface (*Cullen, 2008*). Analogous to the degradation of iRhom2 in iTAP KO cells, loss of Snx17, which binds to the cytoplasmic tail of β1 integrins, results in a failure in their endocytic recycling, leading to their degradation in lysosomes (*Böttcher et al., 2012*). Notably, the iRhom2 cytoplasmic tail contains two motifs, NxxY and NPxY (*Figure 8—figure supplement 1B*) that are the consensus endocytic signals recognized by a subset of FERM-domain containing sorting nexins, involved in endocytic recycling. However, our preliminary experiments in which we have mutated those motifs to alanines (AAAA), show that they appear not to be required for iTAP/iRhom2 binding (*Figure 8—figure supplement 1C*). Moreover, although sorting nexins are intimately connected with the endocytic/recycling machinery, our preliminary experiments detect no obvious colocalization of iTAP with early endosomes (*Figure 8—figure supplement 1D*). Endocytic sorting is however only one theme within the wider FERM biology. Future studies will be required to clarify the relationship between iTAP and the trafficking machinery, to map the vesicular itinerary taken by iRhom/TACE complexes, and to establish the precise basis of the mis-sorting defect in iTAP-null cells.

It will also be interesting to reconcile the role of iTAP in the control of iRhom/TACE stability, versus that of PACS-2, which binds directly to TACE (*Dombernowsky et al., 2015*). iTAP and PACS-2 both impact on TACE stability, but iTAP can presumably only influence TACE stability indirectly via iRhoms. This is relevant because TACE stimulants trigger detachment of TACE from iRhom2 on the cell surface (*Grieve et al., 2017*), a mechanism important for facilitating access of TACE to its substrates (*Cavadas et al., 2017*). As iRhom and TACE are uncoupled at a crucial stage during signaling, their degradative fates could also be separated, leaving open the possibility that iTAP and PACS-2 may govern different stages in TACE's trafficking lifecycle.

The TNF (and EGFR) pathway(s) are very stringently regulated by positive and negative feedback (*Avraham and Yarden, 2011*; *Wallach, 2016*; *Vereecke et al., 2009*). Considering the significant

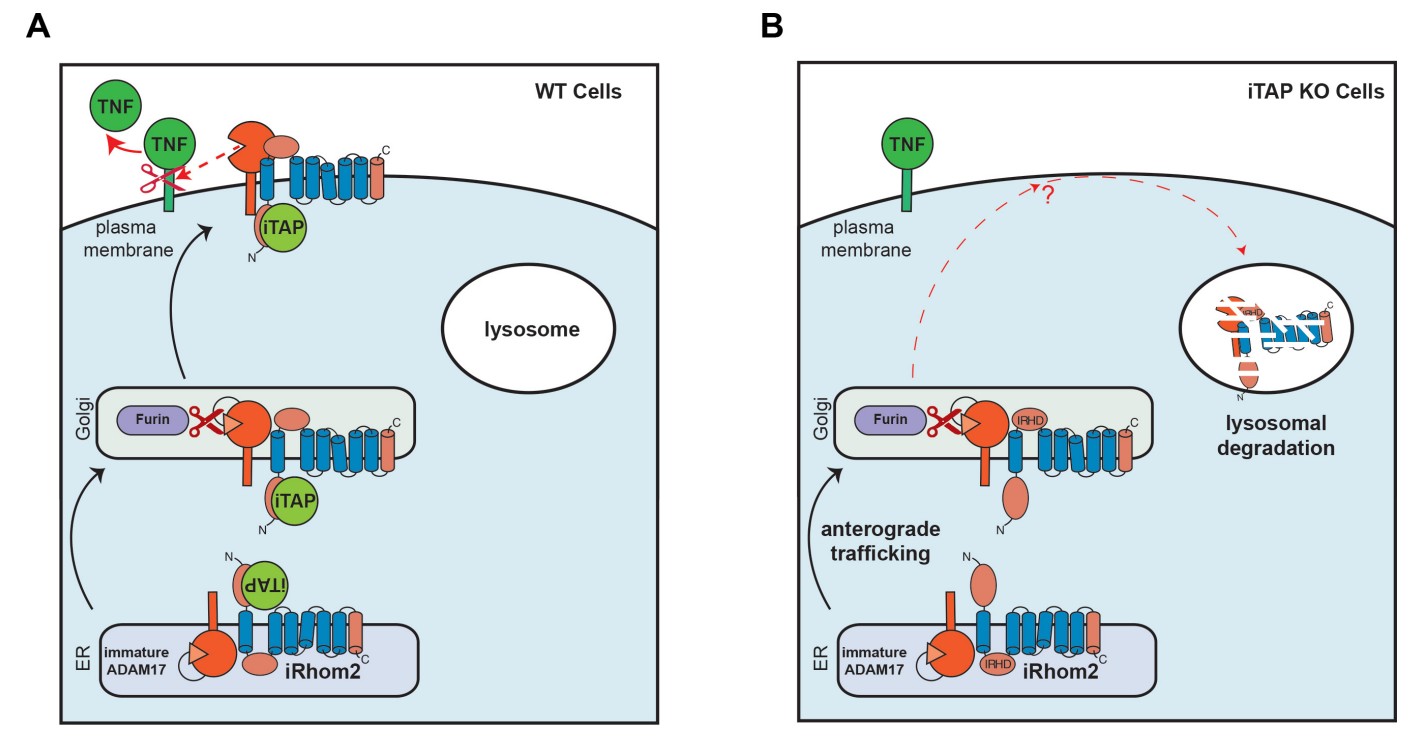

**Figure 8.** Schematic model showing regulation of the cell surface stability of the sheddase complex by iTAP. (**A**). In WT cells the iRhom2/TACE sheddase complex successfully transits from the ER to the Golgi apparatus, where TACE undergoes maturation (prodomain removal). The sheddase complex then traffics to the cell surface, where TACE cleaves it substrates (e.g. TNF, EGFR ligands), enabling their release for signaling. iTAP, which loads onto the sheddase complex in the ER, remains associated with the sheddase complex and ensures the stability of the complex on the cell surface, promoting the cleavage of TACE substrates. (**B**). By contrast, in iTAP KO cells, the sheddase complex is aberrantly sorted to the lysosome, where iRhom2 and mature TACE are degraded. As a result, no TACE substrates are released for signaling. The dotted arrows indicate a putative itinerary taken by the sheddase complex in iTAP KO cells. The sheddase complex may be destabilized on the cell surface: aberrantly targeted for endocytosis and shunted to the lysosome. Alternatively, the sheddase complex may be endocytosed from the cell surface at the normal rate, but loss of iTAP may result in a defect in recycling the complex back to the cell surface, favouring delivery to the lysosome.
DOI: https://doi.org/10.7554/eLife.35032.030

The following figure supplement is available for figure 8:

**Figure supplement 1.** iTAP does not bind to features commonly recognized by FERM domain proteins.
DOI: https://doi.org/10.7554/eLife.35032.031

impact that iTAP has on TACE biology, it is tempting to speculate that feedback control over the signaling pathways that culminate in TNF or EGFR ligand release, could be governed by controlling the interaction between iTAP and iRhom, or by modulating the stability of iTAP itself. Our preliminary experiments suggest that the phosphorylation of iRhom2 at residues required for the stimulation of TACE activity (*Cavadas et al., 2017*) does not appear to influence iTAP binding (data not shown), but future studies are required to investigate more widely the possibility of iTAP regulation by stimuli relevant to TACE biology.

In addition to our extensive evidence indicating the requirement for iTAP for normal TACE function in human and mouse cells, we show that mature TACE levels are dramatically depleted in tissues from iTAP KO mice, including macrophages (*Figure 7D*). This reinforces the notion that iTAP is an important physiological regulator of the sheddase complex at the organismal level, in multiple tissues. Notably, whereas ADAM17 homozygous mutant mice exhibit perinatal lethality (*Peschon et al., 1998*), iTAP KO mice are born at near-normal mendelian ratios (*Table 2*), reach adulthood, appear superficially healthy and are fertile. These data indicate that although loss of iTAP indeed has a profound impact on the levels of mature TACE, the fraction remaining of mature TACE is sufficient to ensure normal mouse development.

**Table 2.** Mendelian ratios of embryos isolated at embryonic day 14.5 (E14.5), or pups at (P1) post-partum obtained from crosses obtained between iTAP heterozygous mice.

| Developmental stage | iTAP +/+ | iTAP +/- | iTAP -/- | Total no. of animals |
|---|---|---|---|---|
| E14.5 | 10 | 17 | 4 | 31 |
| % | 32.2 | 54.8 | 12.9 | |
| Ratio | 1 | 1,7 | 0,4 | |
| P1 | 54 | 116 | 50 | 220 |
| % | 24.55 | 52.73 | 22.73 | |
| Ratio | 1 | 2.14 | 0.93 | |

DOI: https://doi.org/10.7554/eLife.35032.032

Several hypomorphic TACE mutant mice have been studied, including *ADAM17^{ex/ex}* mice, which were generated by the insertion of a new exon containing an in-frame stop codon, flanked by weak splice donor/acceptor sites inside the *Adam17* (TACE) locus (*Chalaris et al., 2010*). 95% of the TACE mRNA produced contains the mutant exon, resulting in a dramatic reduction in TACE levels (*Chalaris et al., 2010*). Notably, these animals are born at normal mendelian ratios but are highly susceptible when challenged to an experimental model of colitis (*Chalaris et al., 2010*). This suggests that while traces of TACE can mitigate against lethality, they are not sufficient to prevent disease challenge. Hence, it will be important to dissect fully, in future, the organismal role of iTAP, particularly within the context of disease.

Inhibiting TACE activity has been the subject of considerable pharmaceutical interest for decades, but attempts have failed, often because of cytotoxicity caused by unintended collateral targeting of ADAMs and matrix metalloproteases, that share active site architectures related to TACE (*Murumkar et al., 2010*).

As iTAP has no apparent impact on other ADAMs, the blockade of the iRhom:iTAP interaction may be an interesting potential therapeutic approach to attenuate TACE activity during disease. Such an approach would obviate the concern of collateral targeting of other metalloproteases. Although iTAP ablation at the cellular level has a potent impact on TACE substrate cleavage, at the organismal level the impact is significantly less severe than the lethal phenotype of TACE KO mice (*Peschon et al., 1998*). This implies that it may be possible to target iTAP to reduce TACE activity sufficiently to achieve a therapeutic impact in diseased tissues, without impinging on the normal physiological roles of TACE, which are sustained with minimal TACE levels in *ADAM17^{ex/ex}* mutants (*Chalaris et al., 2010*) and presumably our iTAP KO mice.

## Materials and methods

### Key resources table

| Reagent type (species) or resource | Designation | Source or reference | Identifiers | Additional information |
|---|---|---|---|---|
| Gene (*Homo sapiens*) | *FRMD8*; iTAP, human | n/a | ENSG00000126391 | |
| Gene (*Mus musculus*) | *Frmd8*; iTAP, mouse | n/a | ENSMUSG00000024816 | |
| Strain, strain background (*Mm*; C57BL/6) | wild type; WT mice | Jackson Laboratories | RRID:IMSR_JAX:000664 | |
| Genetic reagent (*Mm*,C57BL/6) | iTAP KO mice | this paper | n/a | Generated by CRISPR (more details in M and M section and *Figure 7—figure supplement 1*) |
| Cell line (*Hs*) | HEK 293ET | 10.1038/sj.emboj.7601743 | RRID:CVCL_6996 | |
| Cell line (*Hs*) | HeLa | ATCC | ATCC CCL-2; RRID:CVCL_0030 | |
| Cell line (*Mm*) | RAW 264.7 | Sigma | 91062702; RRID:CVCL_0493 | |
| Cell line (*Mm*) | L929 | ATCC | ATCC CCL-1; RRID:CVCL_0462 | |

*Continued on next page*

*Continued*

| Reagent type (species) or resource | Designation | Source or reference | Identifiers | Additional information |
|---|---|---|---|---|
| Cell line (*Hs*) | HEK 293ET iTAP KO | this paper | n/a | Generated by CRISPR (more details in M and M section) |
| Cell line (*Mm*) | RAW 264.7 iTAP KO | this paper | n/a | Generated by CRISPR (more details in M and M section) |
| Cell line (*Mm*) | L929 iTAP KO | this paper | n/a | Generated by CRISPR (more details in M and M section) |
| Cell line (*Mm*) | MEF iTAP KO | this paper | n/a | Isolated from iTAP KO mouse embryos |
| Cell line (*Mm*) | DKO MEF | 10.1038/embor.2013.128. | n/a | Isolated from iRhom1/2 DKO mouse embryos |
| Biological sample (*Mm*) | tissues from WT and KO mice | other | n/a | Isolated with standard techniques from the mice described here |
| Antibody | anti-TACE; TACE Ab 318 | https://doi.org/10.1016/j.jim.2011.06.015 | n/a | |
| Antibody | anti-TACE for Immunoprecipitation | R and D systems | R and D 9301 | |
| Antibody | anti-TACE | abcam | Ab39162; RRID:AB_722565 | Ab39163 |
| Antibody | anti-P97 | Thermo-Fisher | MA1-21412; RRID:AB_557663 | |
| Antibody | anti-TACE | R and D systems | 9301; RRID:AB_2223551 | |
| Antibody | anti-TACE | 10.1016/j.jim.2011.06.015 | Ab318 | monoclonal anti-TACE antibody (*Trad et al., 2011*) was used for the detection of deglycosylated TACE when no conA was used |
| Antibody | anti-tubulin | IGC antibody facility | Clone YL1/2; RRID:AB_793541 | |
| Antibody | anti-HA-HRP | Roche | 3F10; RRID:AB_2314622 | |
| Antibody | anti-HA | Biolegend | 901501; RRID:AB_291259 | |
| Antibody | anti-V5-HRP | Life Technologies | R961-25 | |
| Antibody | anti-Frmd8 | Abnova | 157H00083786-B01P; RRID:AB_1573641 | |
| Antibody | anti-Transferrin Receptor | Life Technologies | 13–6800; RRID:AB_2533029 | |
| Antibody | anti-Flag-HRP | Sigma | A8592; RRID:AB_439702 | |
| Antibody | anti-GAPDH | Cell signalling Technology | 2118; RRID:AB_561053 | |
| Antibody | anti-actin | Abcam | Ab8227; RRID:AB_2305186 | |
| Antibody | anti-iRhom2 | 10.1126/science.1214400 | n/a | Anti-iRhom2 polyclonal antibodies specific to the mouse iRhom2 N-terminus (amino acids 1–373) or raised against the iRhom homology domain, were previously described (*Adrain et al., 2012*) |
| Antibody | Anti-HA magnetic beads | Pierce | 88836, Pierce | |
| Antibody | Mouse anti-GFP | IGC antibody Facility | clone 19F7 | Used as IP negative control in *Figure 3—figure supplement 1* |
| Antibody | Anti-Flag M2 affinity gel | Sigma | A220 | |
| Antibody | MagnaBind Goat Anti-Rabbit IgG Beads | Thermo Scientific | 21356 | |

*Continued on next page*

*Continued*

| Reagent type (species) or resource | Designation | Source or reference | Identifiers | Additional information |
|---|---|---|---|---|
| Antibody | MagnaBind Goat Anti-Mouse IgG Beads | Thermo Scientific | 21354 | |
| Antibody | mouse anti-LAMP2 | DSHB | Clone H4B4; RRID:AB_2134755 | |
| Antibody | rabbit anti-EEA1 | Cell signalling Technology | 3288; RRID:AB_2096811 | |
| Antibody | rabbit anti-Strumpellin/WASHC5 | Santa Cruz | SC87442; RRID:AB_2234159 | |
| Transfected construct (*Hs*) | h iRhom1-HA (plasmid) | this paper | n/a | |
| Transfected construct (*Hs*) | h iRhom1 Nterm HA (plasmid) | this paper | n/a | |
| Transfected construct (*Mm*) | Rhbdd2-HA (plasmid) | this paper | n/a | |
| Transfected construct (*Hs*) | RHBDD3-HA (plasmid) | this paper | n/a | |
| Transfected construct (*Mm*) | Ubac2-HA (plasmid) | this paper | n/a | |
| Recombinant DNA reagent | pM6P.blast-GFP (plasmid) | Felix Randow | n/a | |
| Recombinant DNA reagent | pLEX.blast (plasmid) | this paper | n/a | Details in M and M |
| Transfected construct (*Mm*) | iRhom2-Cherry (plasmid) | doi:10.1038/ni.3510 | n/a | |
| Transfected construct (*Hs*) | ADAM33-V5 (plasmid) | 10.1038/embor.2013.128. | n/a | |
| Transfected construct (*Hs*) | ADAM19-V5 (plasmid) | 10.1038/embor.2013.128. | n/a | |
| Transfected construct (*Hs*) | ADAM22-V5 (plasmid) | 10.1038/embor.2013.128. | n/a | |
| Transfected construct (*Hs*) | ADAM8-V5 (plasmid) | 10.1038/embor.2013.128. | n/a | |
| Transfected construct | TNF-FLAG (plasmid) | this paper | n/a | |
| Transfected construct (*Hs*) | STING-FLAG (plasmid) | doi:10.1038/ni.3510 | n/a | |
| Transfected construct (*Hs*) | LAMP1-mCherry (plasmid) | this paper | n/a | Details in M and M |
| Transfected construct (*Mm*) | GFP-iRhom2 (plasmid) | this paper | n/a | Details in M and M |
| Transfected construct (*Mm*) | mCherry-iRhom2 (plasmid) | this paper | n/a | Details in M and M |
| Transfected construct (*Mm*) | TACE-GFP (plasmid) | this paper | n/a | Details in M and M |
| Transfected construct (*Mm*) | TACE-TagRFP (plasmid) | this paper | n/a | Details in M and M |
| Transfected construct (*Mm*) | mouse iTAP-mCherry (plasmid) | this paper | n/a | Details in M and M |
| Transfected construct (*Hs*) | human iTAP-GFP (plasmid) | this paper | n/a | Details in M and M |
| Transfected construct (*Mm*) | iRhom2 NPAY > AAAA (plasmid) | this paper | n/a | Quick change-based mutagenesis on iRhom2-HA plasmid described in M and M |

*Continued on next page*

*Continued*

| Reagent type (species) or resource | Designation | Source or reference | Identifiers | Additional information |
|---|---|---|---|---|
| Transfected construct (*Mm*) | iRhom2 NRSY > AAAA (plasmid) | this paper | n/a | Quick change-based mutagenesis on iRhom2-HA plasmid described in M and M |
| Transfected construct (*Mm*) | iRhom2 Double NxxY > AAAA (plasmid) | this paper | n/a | Quick change-based mutagenesis on iRhom2-HA plasmid described in M and M |
| Transfected construct (*Hs*) | SREBP2-FLAG (plasmid) | DOI: 10.1083/jcb.201305076 | n/a | |
| Recombinant DNA reagent | EGF-AP; EGF (plasmid) | doi: 10.1083/jcb.200307137 | n/a | |
| Recombinant DNA reagent | Betacellulin-AP; BTC-AP; BTC (plasmid) | doi: 10.1083/jcb.200307137 | n/a | |
| Recombinant DNA reagent | TNF-AP; TNF (plasmid) | doi: 10.1083/jcb.200307137 | n/a | |
| Recombinant DNA reagent | TGFα-AP; TGFα (plasmid) | doi: 10.1083/jcb.200307137 | n/a | |
| Recombinant DNA reagent | HB-EGF-AP; HB-EGF (plasmid) | doi: 10.1083/jcb.200307137 | n/a | |
| Recombinant DNA reagent | Epiregulin-AP; EPIREG-AP; EPIREG (plasmid) | doi: 10.1083/jcb.200307137 | n/a | |
| Recombinant DNA reagent | Amphiregulin-AP; AREG-AP; AREG (plasmid) | doi: 10.1083/jcb.200307137 | n/a | |
| Recombinant DNA reagent | secreted luciferase | 10.1038/embor.2013.128. | n/a | |
| Recombinant DNA reagent | pLenti-Crispr version2 (plasmid) | Addgene | 52961 | |
| Recombinant DNA reagent | pLentiCas9-Blast(plasmid) | Addgene | 52962 | |
| Recombinant DNA reagent | pLentiGuide-Puro(plasmid) | Addgene | 52963 | |
| Recombinant DNA reagent | pgRNAbasic(plasmid) | doi: 10.1242/dev.133074 | n/a | |
| Recombinant DNA reagent | pT7-Cas9(plasmid) | doi: 10.1242/dev.133074 | n/a | |
| Sequence-based reagent (siRNA; *Hs*) | iTAP oligo #1 | Santa Cruz | sc-96500 | |
| Sequence-based reagent (siRNA; *Hs*) | iTAP oligo #2 | GE Dharmacon | M-018955-01-0005 | |
| Sequence-based reagent (siRNA; *Hs*) | TACE oligo #1 | Santa Cruz | sc-36604 | |
| Sequence-based reagent (siRNA; *Hs*) | TACE oligo #2 | GE Dharmacon | M-003453-01-0005 | |
| Sequence-based reagent (CRISPR Guide sequence; *Hs*; cell lines) | gRNA targeting exon 1 | this paper | n/a | 5'-GCCCCGCTGAGCGATCCCAC-3' |
| Sequence-based reagent (CRISPR Guide sequence; *Hs*; cell lines) | gRNA targeting exon 4 | this paper | n/a | 5'-ACGTGTTCTTCCCAAAGCGG-3' |
| Sequence-based reagent (CRISPR Guide sequence; *Hs*; cell lines) | gRNA targeting exon 2 | this paper | n/a | 5'-TGACGTGCTGGTATACCTAG-3' |
| Sequence-based reagent (CRISPR Guide sequence; *Hs*; cell lines) | gRNA targeting exon 6 | this paper | n/a | 5'-GGCACTTGAGGAGATAGGCG-3' |

*Continued on next page*

Continued

| Reagent type (species) or resource | Designation | Source or reference | Identifiers | Additional information |
|---|---|---|---|---|
| Sequence-based reagent (CRISPR Guide sequence; *Mm*; cell lines) | gRNA targeting first exon | this paper | n/a | 5'-TTCGGTGGGACCGCTCCGCA-3' |
| Sequence-based reagent (CRISPR Guide sequence; *Mm*; cell lines) | gRNA targeting second exon | this paper | n/a | 5'-GCACTACTGTATCATCCGCC-3' |
| Commercial assay or kit | fluorogenic TACE substrate peptide | ENZO Life Sciences | BML-P235-0001 | |
| Commercial assay or kit | 1-step PNPP Substrate | Thermo Fisher | PIER37621 | |
| Commercial assay or kit | Fugene 6 | Promega | 2691-SC | |
| Commercial assay or kit | Endoglycosidase; Endo-Hf | NEB | 174P0703 | |
| Commercial assay or kit | PNGase F | NEB | 174P0704 | |
| Commercial assay or kit | Concanavalin A Agarose; conA | G-biosciences | 786–216 | |
| Commercial assay or kit | MEGAshortscript T7 Kit | Thermo Fisher | AM1354 | |
| Commercial assay or kit | MEGAclear kit | Thermo Fisher | AM1908 | |
| Commercial assay or kit | Gibson Assembly Master Mix | New England Biolabs | 174E2611 | |
| Commercial assay or kit | KOD Hotstart DNA polymerase | Novagen | 71086–5 | |
| Commercial assay or kit | TOPO TA cloning kit for sequencing | Invitrogen | 450030 | |
| Commercial assay or kit | mMESSAGE mMACHINE T7 Ultra Kit | Thermo Fisher | AM1345 | |
| Commercial assay or kit | human TNF ELISA kit | R and D Systems | DY210 | |
| Commercial assay or kit | human IL-6 ELISA kit | R and D systems | DY-206 | |
| Commercial assay or kit | human IL-8 ELISA kit | R and D systems | DY-208 | |
| Chemical compound, drug | Polyethylenimine; PEI | Sigma | 408727 | |
| Chemical compound, drug | 1,10-phenanthroline | Sigma | 131377 | |
| Chemical compound, drug | Bafilomycin A1 | Santa cruz | 201550 | |
| Chemical compound, drug | Chloroquine | Sigma | C6628 | |
| Chemical compound, drug | Ammonium chloride | Acros | 10676052 | |
| Chemical compound, drug | Ionomycin | Cayman | CAYM11932-5 | |
| Chemical compound, drug, (*E.coli 055:B5*) | Lipopolysaccharide; LPS | Santa cruz | (sc-221855A) | |
| Chemical compound, drug | Phorbol 12-myristate 13-acetate; PMA | Sigma | P1585 | |

*Continued*

| Reagent type (species) or resource | Designation | Source or reference | Identifiers | Additional information |
|---|---|---|---|---|
| Chemical compound, drug | Cycloheximide; CHX | Santa cruz | sc-3508 | |
| Chemical compound, drug | Dithiobis (succinimidyl propionate); DSP | Pierce | 10731945 | |
| Chemical compound, drug | Sulfo-NHS-LC-Biotin | Thermo Scientific, | 21335 | |
| Other | Lenti-X Concentrator | Clontech | Clonetech: 631231 | |
| Other | NeutrAvidin agarose; Neutravidin resin | Thermo Fisher | 11885835 | |
| Other | NuPAGE Novex 4–12% Bit-Tris Protein Gels 1.0 mm | Novex; Life Technologies | NP0322BOX | |
| Other | Opti-MEM I Reduced Serum Medium, GlutaMAX | Life Technologies | 51985–026 | |
| Software, algorithm | Fiji | PMID: 22743772 | RRID:SCR_002285 | |
| Software, algorithm | Illustrator | Adobe | Adobe Creative Suite | |
| Software, algorithm | GraphPad Prism | GraphPad Software, Inc. | | |
| Software, algorithm | Volocity | PerkinElmer | | |
| Software, algorithm | Geneious | Biomatters Ltd. | Javaversion 1.8. 0_71-b15 | |
| Software, algorithm | perkin elmer 2030 workstation | PerkinElmer | | |

## Plasmids

C-terminally triple HA-tagged versions of human iRhom1, the cytoplasmic N-terminus of iRhom1 (amino acids 1–404), mouse iRhom2, mouse Rhbdd2, human RHBDD3 and mouse Ubac2 were cloned into the lentiviral expression plasmid pLEX-MCS, using Gibson cloning. These plasmids were used only for the respective mass spectrometry experiments. C-terminally triple FLAG-tagged iTAP and mouse iRhom2-HA were cloned into the retroviral pM6P vector (a kind gift of Felix Randow) using Gibson assembly. The N-terminal truncations of iRhom2 used in *Figure 2* were cloned into a modified version of the lentiviral expression vector pLEX-MCS in which the puromycin resistance cassette was replaced with a blasticidin resistance gene. The packaging vectors for the production of retrovirus or lentivirus were described previously (*Cavadas et al., 2017*). A Cherry-tagged iRhom2 plasmid previously described (*Luo et al., 2016*) was used only for the experiments in *Figure 2—figure supplement 1*. LAMP1-mCherry was subcloned from Addgene plasmid #45147 (a gift from Amy Palmer) into pM6P vector with zeocin resistance (prepared by replacing the blasticidin resistance gene in the original vector with zeocin resistance). Mouse eGFP-iRhom2 and mouse TACE-GFP were cloned using Gibson assembly into pM6P.HisD vector. Mouse TACE-TagRFP was cloned using Gibson assembly into pM6P.Blast, using Addgene plasmid #42635 (a gift from Silvia Corvera) as a source of TagRFP DNA and a mTACE-GFP plasmid from Jürgen Scheller as a source of mTACE cDNA. Murine iTAP-mCherry (iTAP cDNA from Origene Technologies) and mCherry-iRhom2 were cloned using Gibson assembly into pM6P.Blast. Human iTAP was inserted with standard cloning techniques into the eGFP containing pIC111 vector (pIC111 is gift from Iain Cheeseman and Arshad Desai; Addgene plasmid # 44435). Alkaline phosphatase-tagged TACE substrates, a gift of Shigeki Higashiyama were described previously (*Sahin et al., 2004*). V5-tagged ADAM expression plasmids and secreted luciferase construct were described previously (*Christova et al., 2013*). CRISPR plasmids are described below. Human Tumor Necrosis Factor (TNF) containing an N-terminal FLAG tag, was cloned into pCR3 by standard techniques. Flag-tagged SREBP2 was a gift of Larry Gerace and STING-FLAG a gift of Lei Jing. In *Figure 8—figure supplement 1C*, iRhom2-HA in a modified version of pEGFP-N1/non EGFP was used as a template for Quick-Change mutagenesis resulting in the constructs iRhom2 NPAY >AAAA, iRhom2 NRSY >AAAA and iRhom2 Double NxxY >AAAA.

## Cell culture

Our work involved the use of cell lines. Routine testing for mycoplasma revealed mycoplasma negative status. None of our lines are identified on the list of commonly misidentified cell lines provided by ILCAC. We have noted the sources (e.g. ATCC accession number) of our lines in the Key resources table. HEK 293ET, RAW 264.7, L929, MEF and HeLa cell lines were maintained under standard conditions in Dulbecco's Modified Eagle Medium (DMEM)-high glucose supplemented with fetal bovine serum. Bone Marrow Derived Macrophages (BMDM) were isolated from 8 week old mice and cultured as previously described (*Adrain et al., 2012*). Embryonic fibroblasts were generated from E14.5 embryos and immortalized using lentiviral transduction of SV40 virus large T antigen.

## Cytokine secretion in isolated human monocytes

Primary human peripheral blood mononuclear cells (PBMC) were purified from donor whole blood using the Ficoll-Hypaque gradient method as described previously (*Henry and Martin, 2017*). After overnight plastic adherence in heat-inactivated serum containing medium, non-adherent cells were removed and remaining cells were washed three times in PBS. Macrophage differentiation was induced using recombinant human macrophage-colony stimulating factor (M-CSF, 100 ng/ml) over five-seven days during cell culture in RPMI supplemented with 10% FCS. Primary human macrophages ($5 \times 10^5$) were nucleofected with 200 nM of each siRNA (control NS oligo, MWG Eurofins - 5'- guuccugagccuggacuac −3'; iTAP oligo #1, Santa Cruz - catalog code sc-96500; iTAP oligo #2, GE Dharmacon - M-018955-01-0005; TACE oligo #1, Santa Cruz - sc-36604; TACE oligo #2, GE Dharmacon - M-003453-01-0005) in nucleofection buffer (5 mM KCl, 15 mM MgCl$_2$, 20 mM HEPES, 150 mM Na$_2$HPO$_4$ [pH 7.2]) using Amaxa Nucleofector (program Y-010). Cells were plated in 6-well plates ($2 \times 10^5$ cells/well) or in 24-well plates ($1 \times 10^5$ cells/well) and 48 hr after nucleofection were stimulated with lipopolysaccharide (LPS). After 18 hr, cell culture supernatants were collected and clarified by centrifugation for 5 min at 800 x g. Cytokines and chemokine concentrations were measured from clarified cell culture supernatants using specific ELISA kits obtained from R and D Biotechne systems (human TNF – DY210; human IL-6 – DY206; IL-8 – DY208).

## Retroviral transduction

HEK 293ET cells ($1 \times 10^6$) were transfected with pCL (-Eco, or 10A1) packaging plasmids (*Naviaux et al., 1996*) plus pM6P.BLAST empty vector (kind gift of F. Randow, Cambridge, UK) or pM6P containing the cDNA of human or mouse iTAP or mouse iRhom2. WT or iTAP KO HEK 293ET cells were transduced with the viral supernatant supplemented with polybrene 8 µg/mL, and selected with blasticidin (8 µg/mL) to generate stable cell lines. To transduce RAW 264.7 or L929, lentiviruses were prepared and concentrated as follows: HEK 293FT cells ($24 \times 10^6$) were transfected with pMD-VSVG envelope plasmid, psPAX2 helper plasmid and pLentiCas9-blast or empty vector pLentiGuide-puro or pLentiGuide-puro containing iTAP targeted gDNAs. The viral supernatants were concentrated 300-fold using ultracentrifugation (90,000 g) at 4°C for 4 hr, followed by re-suspension in 0.1% BSA in PBS. Cells were transduced with the concentrated virus, supplemented with 8 µg/ml polybrene.

## Generation of iTAP KO cells via CRISPR

For CRISPR-mediated knockout of iTAP in human cells, gRNAs targeting exons common to all transcripts: the first coding exon 5'-GCCCCGCTGAGCGATCCCAC-3' or coding exon 4 of *FRMD8* (iTAP) 5'-ACGTGTTCTTCCCAAAGCGG-3' were cloned into pLentiCRISPR v2 (Addgene plasmid # 52961), a gift from Feng Zhang. For the ablation of iTAP in human cells, HEK 293ET cells ($2.5 \times 10^4$/ cm$^2$) were transfected using Fugene with pLentiCRISPR v2 empty vector, or either of the pLentiCRISPR-derived sgRNA plasmids described above. The next day, the cells were selected with puromycin (8 µg/mL) for 3 days until mock transfected cells were eliminated. Cells were expanded and single-cell sorted by FACS or serial dilutions on 10 cm culture plates. To screen for the presence of indels in clones, genomic DNA was extracted from each clone and a 200 bp region flanking the site targeted by the gRNA was amplified for exon 1 (forward = 5'-CCTCCAGCCCCCCATCCCTGGCTC-3'; reverse = 5'-GCCAGAGCTACTTCTCCAGGGCTGGGG-3') or exon 4 (forward = 5'-TCGGGAGAGGGGAGGGCTAAGCAG-3'; reverse = 5'-GGGCAAGGTGCGAATGTCCAGGGGTC-3'). Clones with mutant alleles were selected and the original PCR fragments amplified were isolated and

sequenced via TOPO TA cloning. The selected clones 'KO A' and 'KO B' which contain indels in all alleles of *FRMD8*, were then confirmed for loss of iTAP at the protein level by immunoprecipitation and subsequent western blot with an anti-FRMD8 antibody. For the ablation of iTAP in HeLa cells, an alternative approach was used: HeLas were transiently transfected with a pool of 3 gRNA plasmids (PX330, Zhang lab, Addgene 42230 [*Cong et al., 2013*]) (TGACGTGCTGGTATACCTAG; GGAACGTGTTCTTCCCAAAG; GGCACTTGAGGAGATAGGCG) specifically targeting exons 2,4 and 6 of the human FRMD8 gene respectively, in conjunction with a plasmid encoding puromycin resistance (pEGFP-C1 from Clontech expressing a GFP-tagged puromycin N-acetyl-transferase). After puromycin selection, the efficient knockout of iTAP in the bulk population was confirmed by immunoblotting the lysates with iTAP antibodies, and by the significant depletion of mature TACE (*Figure 6—figure supplement 1*).

To ablate iTAP in mouse cells, gRNAs targeting the first coding exon (5'-TTCGGTGGGACCGCTCCGCA-3') or second coding exon (5'-GCACTACTGTATCATCCGCC-3') were cloned into pLenti-Guide-Puro (Addgene plasmid # 52963) and used in combination with pLentiCas9-Blast (Addgene plasmid # 52962); both gifts of Feng Zhang. For transfection of the gRNAs, $2 \times 10^5$ L929 or $5 \times 10^5$ RAW 264.7 cells were transduced with 40 µl (RAW 264.7) or 20 µl (L929) of 300-fold concentrated lentivirus encoding pLentiCas9-Blast (Addgene #52962) and selected with blasticidin (4 µg/mL, RAW 264.7 and 8 µg/mL, L929). The Cas9-expressing lines were then transduced with the pLentiGuide-Puro sgRNA plasmds targeting the first or second coding exons of mouse *Frmd8* (iTAP). Following selection with puromycin (4 µg/mL, RAW 264.7 and 7 µg/mL L929) the cells were single clone sorted by FACS. To screen for iTAP KO clones, genomic DNA was extracted from each clone and PCR used to amplify a 200 bp region flanking the guide sequence (exon 1: forward = 5'TTGAGAGCTTGAGGAGACCA-3'; reverse 5'-CAGGCTGGAACCAAAGAGTTC-3'; exon 2: forward = 5'-GGAAATGCTGATTGGACCTC-3'; reverse 5'-CCTGCTGCCAGACCTTACCC-3'). Clones with mutant alleles were identified as described for human cells.

## Experiments with mice

Experiments with mice were performed in accordance with protocols approved by the Ethics Committee of the Instituto Gulbenkian de Ciência and the Portuguese National Entity (DGAV-Direção Geral de Alimentação e Veterinária) and with the Portuguese (Decreto-Lei no. 113/2013) and European (directive 2010/63/EU) legislation related to housing, husbandry, and animal welfare.

Generation of iTAP mutant mice iTAP mutant mice were generated via CRISPR/Cas9 as previously described (*Wang et al., 2013*; *Casaca et al., 2016*). In brief, two gRNA´s (5'-CAGCCGAGTGCAGATCGGGT-3' and 5'-GTGGCGGACTCAGAAATCAA-3') were designed to introduce a deletion of the first coding exon (exon 2) of the mouse *Frmd8* gene. Oligos encoding the gRNA were inserted into the plasmid pgRNAbasic (*Casaca et al., 2016*), which contains a T7 promoter. The linearized vector was used as template for the production of sgRNAs, produced by *in vitro* transcription using the MEGAshortscript T7 Kit (Life Technologies). RNA was cleaned using the MEGAclear kit (AM1908, Life Technologies). Cas9 mRNA was produced by *in vitro* transcription using the mMESSAGE mMACHINE T7 Ultra Kit (Life Technologies) and plasmid pT7-Cas9 as a template (*Casaca et al., 2016*). Cas9 mRNA (10 ng/ml), plus the sgRNAs (10 ng/ml) were injected into the pronuclei of fertilized C57 BL/6 oocytes using standard procedures (*Hogan et al., 1994*). Deletions were assessed by PCR from tail genomic DNA using primers 5'-CCCGACTTGTTTGGCCATTTC-3' or 5'-CGGGGCCTCGGGTTTG-3' (forward) and 5'-TGGGACAAAGGAAGTGGTGCC-3' (reverse). The deletion was confirmed by direct sequencing and TOPO-cloning followed by sequencing. These primers (along with 5'-ACTTTCACCCTACACATTTG-3' 5'-AGTCCGCCACATCTAAAC-3' for better amplification of WT alleles) were also used for genotyping mice and embryos of the iTAP KO line.

## Immunostaining and fluorescence microscopy

HeLa ($5 \times 10^4$ cells/well) were plated on coverslips and transfected with iTAP-GFP (600 ng) or iRhom2-Cherry (600 ng), either alone or in combination. After 24 hr, cell supernatant was removed and cells were washed three times with PBS (2 mL). Cells were fixed with 3% paraformaldehyde for 10 min. Cells were washed again three times with PBS (2 mL) followed by permeabilisation with 0.15% TX100 for 15 min. Cells were blocked with 2% BSA (in PBS, pH 7.2) for 1 hr to reduce non-specific binding of antibodies. Specific primary antibodies against Calnexin (Cell Signaling, C5C9)

and Golgi GM130 (BD, 610823) were diluted 1:100 in 2% BSA. Primary antibodies were incubated for 2 hr at room temperature. Cells were washed three times with PBS (2 mL). Cells were incubated with the relevant rhodamine red-conjugated secondary antibody (Alexa Fluor) diluted 1:1000 in 2% BSA for 1 hr at room temperature. Cells were washed again with PBS, followed by incubation with Hoechst (Sigma) for 10 min. Coverslips were mounted on slides with 5 μ∧ of Slow Fade (Molecular Probes).

For mitotracker staining, cells were transfected with iTAP-GFP (600 ng), as described previously. After 24 hr, cells were treated with Mitotracker-Red (50 nM) for 15 min at 37°C, followed by fixation with 3% paraformaldehyde. Nuclei were stained with Hoechst. Cells were visualised and analysed using a laser scanning confocal microscope (Olympus FV1000) using a 488 nm Argon laser (green fluorescence), a 543 nm HeNe laser (red fluorescence) and a 405 nm LD laser. Confocal images were acquired using Fluroview 1000 V.1 software.

To investigate the lysosomal mis-sorting of mCherry-iRhom2 in iTAP deficient HeLa cells, 1 μg mCherry-iRhom2 or 500 ng mCherry-iRhom2 and 500 ng iTAP-GFP were transfected into parental and iTAP KO HeLa cells using Fugene 6 (Promega). 48 hr post transfection, cells were fixed in 4% PFA in PBS, permeabilized with 0.2% Saponin in PBS, blocked in 0.1% Saponin/1% BSA in PBS and incubated with an antibody against the lysosomal marker LAMP2 and appropriate secondary antibodies. The cells expressing mCherry-iRhom2 and iTAP-GFP were also co-stained under the same conditions with the early endosome marker EEA1 to visualize a potential localization to endosomes.

## Live-cell fluorescence microscopy

iRhom1/iRhom2 DKO MEFs expressing eGFP-miRhom2 alone or with mouse iTAP-mCherry (*Figure 5H*), or WT MEFs stably expressing eGFP-miRhom2, mTACE-GFP, mTACE-TagRFP, or LAMP1-mCherry delivered by retroviral transduction (using pM6P derivatives) in the indicated combinations were plated ($5 \times 10^4$ per well) on 4-chamber glass-bottomed dishes (In Vitro Scientific, D35C4-20-1.5-N) 24 hr prior to imaging, in the presence or absence of 10 μM Chloroquine as indicated. Cells were imaged on a laser scanning confocal microscope Zeiss LSM 780* using the 40x/1.2 M27 W Korr C-Apochromat objective and a 488 or 561 nm excitation wavelength.

## *In vivo* protein cross-linking with Dithiobis Succinimidyl Propionate (DSP)

Cells were washed twice in cold PBS before incubation in 0.2 mg/mL DSP for 45 min. The cross-linker was aspirated off and the cell monolayers were washed three times for 10 min in ice cold PBS containing 50 mM Tris, pH 8.0 to quench any remaining cross-linker. Subsequently the cells were lysed in Triton X-100 lysis buffer (150 mM NaCl, 50 mM Tris-HCl, protease inhibitors, pH 7.4 and 10 mM 1,10-phenanthroline). Post-nuclear supernatants were supplemented to contain 0.1% SDS and 0.25% sodium deoxycholate.

## Co-immunoprecipitations

HEK 293ET cells expressing the indicated plasmids were lysed for 10 min on ice in TX-100 lysis buffer (1% Triton X-100, 150 mM NaCl, 50 mM Tris-HCl, pH 7.4) containing complete protease inhibitor cocktail (Roche), and 10 mM 1,10-phenanthroline (to inhibit TACE autoproteolysis) unless otherwise indicated. Post-nuclear supernatants were pre-cleared with unconjugated magnetic beads or agarose at 4°C for 60 min with rotation, followed by capture on anti-HA magnetic beads or anti-FLAG respectively for 90 min. Beads were washed 3–5 times, for 10 min, at 4°C in the same Triton X-100 lysis buffer supplemented with NaCl to 300 mM. Samples were eluted with 1.5 x SDS-PAGE sample buffer and incubated at 65°C for 15 min before loading.

## Identification of iRhom-interacting proteins

HEK 293ET cells were stably transduced with lentiviruses encoding pLEX empty vector, or pLEX derivatives containing HA-tagged iRhom1, iRhom2, iRhom1 N terminus, Rhbdd2, RHBDD3, Ubac2. Live cells were washed twice with ice-cold PBS, then left untreated or treated with the crosslinker DSP (0.2 mg/mL) as described below. Lysates were clarified, then pre-cleared with irrelevant control antibodies conjugated to magnetic beads for 60 min at 4°C with rotation. After saving 'input' samples, the lysates were incubated with anti-HA resin for 90 min at 4°C with rotation. Subsequently, the

precipitated beads were washed four times in the respective buffers indicated above. One quarter of the beads were reserved for SDS-PAGE analysis, whereas three-quarters of the precipitated beads were resuspended in UREA buffer (8M Urea, 4% CHAPS, 100 mM DTT, 0.05% SDS). For MS analysis, immunoprecipitates were enzymatically digested on 3 kD MWCO filters (Pall Austria Filter GmbH) using an adaption of the FASP protocol as described previously (*Bileck et al., 2014*; *Slany et al., 2016*). After pre-concentration of the samples, protein reduction and alkylation was performed, then trypsin was added and incubated at 37°C for 18 hr. The digested peptide samples were dried and stored at −20°C then later reconstituted in 5 µl 30% formic acid (FA) containing 10 fmol each of 4 synthetic standard peptides and further dilution with 40 µl mobile phase A (98% H$_2$O, 2% ACN, 0.1% FA), as described previously (*Bileck et al., 2014*; *Wiśniewski et al., 2009*). LC-MS/MS analyses were performed using a Dionex Ultimate 3000 nano LC-system coupled to a QExactive orbitrap mass spectrometer equipped with a nanospray ion source (Thermo Fisher Scientific). For LC-MS/MS analysis, 5 µl of the peptide solution were loaded and pre-concentrated on a 2 cm x 75 µm C18 Pepmap100 pre-column (Thermo Fisher Scientific) at a flow rate of 10 µl/min using mobile phase A. Following this pre-concentration, peptides were eluted from the pre-column to a 50 cm x 75 µm Pepmap100 analytical column (Thermo Fisher Scientific) at a flow rate of 300 nl/min and further separation was achieved using a gradient from 7 to 40% mobile phase B (80% ACN, 20% H$_2$O, 0.1% FA) over 85 min including column washing and equilibration steps. For mass spectrometric analyses, MS scans were accomplished in the range from m/z 400–1400 at a resolution of 70000 (at m/z = 200). Subsequently, data-dependent MS/MS scans of the eight most abundant ions were performed using HCD fragmentation at 30% normalized collision energy and analyzed in the orbitrap at a resolution of 17500 (at m/z = 200). Protein identification was achieved using Proteome Discoverer 1.4 (Thermo Fisher Scientific, Austria) running Mascot 2.5 (Matrix Science). Therefore, raw data were searched against the human proteome in the SwissProt Database (version 11/2015 with 20.193 entries) with a mass tolerance of 50 ppm at the MS1 level and 100 mmu at the MS2 level, allowing for up to two missed cleavages per peptide. Further search criteria included carbamidomethylation as fixed peptide modification and methionine oxidation as well as protein N-terminal acetylation as variable modifications.

## Mass-spectrometry analysis of iTAP-interacting proteins

Lysates from HEK 293ET cells transfected with either empty vector, iTAP-FLAG, TNF-FLAG, STING-FLAG or SREBP2-FLAG and subjected to an immunoprecipitation with anti-FLAG M2 Affinity Gel. The beads were digested with mass spectrometry-grade porcine trypsin (Promega) 10 ng/µl, in 2M urea, 50 mM tris-HCL pH7.5 and 1 mM DTT, overnight at 37°C. The peptides were alkylated with Iodoacetamide (Sigma), desalted using Empore Octadecyl C18 extraction disks and analysed on a Q-Exactive + mass spectrometer coupled to a nano uHPLC (Thermo Fisher). Analysis was performed with MaxQuant 1.5.8.3 software. The abundance of the different interactors was determined with the average of protein peptides detected in each sample.

## Cell surface biotinylation

Biotinylation was performed as previously described for BMDM with small modifications (*Adrain et al., 2012*). RAW 264.7 macrophages or iRhom2-HA HEK 293ET (1.5 × 10$^6$ cells, six well plates) were moved to a cold room (at 4°C), washed with ice-cold PBS pH 8.0 for 10 min, incubated with (1 mg/mL) Sulfo-NHS-LC-Biotin in PBS pH 8.0, according to the manufacturer's instructions. Following quenching with 50 mM Tris in PBS, cells were lysed for 10 min with TX-100 lysis buffer (1,10-phenathroline, protease inhibitors, 50 mM Tris), then biotinylated surface proteins from post-nuclear supernatants were captured on neutravidin agarose resin at 4°C overnight. The resin was washed three times, 10 min, with TX-100 lysis buffer containing 300 mM NaCl at 4°C. Samples were eluted with 1.5 x SDS-PAGE sample buffer and incubated at 65°C for 15 min, before loading.

## Glycoprotein enrichment using concanavalin A

To improve the detection of TACE, cells were lysed in TX-100 lysis buffer supplemented with 1 mM EDTA, 1 mM MnCl$_2$, 1 mM CaCl$_2$ and glycoproteins were captured using Concanavalin A (ConA) Agarose. Beads were washed twice in the same buffer and eluted by heating for 15 min at 65°C in

sample buffer supplemented with 15% sucrose or for 5 min at 95°C in sample buffer supplemented with 15% sucrose and 1x Glycoprotein denaturation buffer (NEB).

### In vitro protein deglycosylation analysis

Post-nuclear lysate supernatants or denatured lysates, are denatured at 65°C for 15 min in the presence of 1x Glycoprotein Denaturing buffer (NEB). Endo-H and PNGase F reactions are set up according to manufacturer's instructions for 1 hr at 37°C.

## Shedding and secretion assays

Shedding assays were performed using previously described plasmids encoding alkaline phosphatase-tagged EGFR ligands: Transforming Growth Factor α (TGFα), Amphiregulin (AREG), Epiregulin (EPIREG), Heparin Binding-Epidermal Growth Factor (HB-EGF), Epidermal Growth Factor (EGF) and Betacellulin (BTC) or Tumor necrosis factor (TNF) (*Sahin et al., 2006*; *Zheng et al., 2002*). HEK 293ET (3 × 105 in six well-plates) were transfected with 1 µg cDNA of AP-substrates and 6 µL PEI. 48 hr later, cells were washed three times with serum free media before incubation for 1 hr in 1 mL Optimem (containing the vehicle of the drug in next step) (for basal shedding), followed by 1 hr with 1 mL Optimem containing 1 µM PMA (Phorbol 12-myristate 13-acetate) or 2.5 µM IO (Ionomycin; for stimulated shedding). Supernatants from each incubation step were collected. Following, the cells were washed in ice-cold PBS three times and lysed in Triton X-100 buffer described previously (unshed material). Supernatants and lysates were centrifuged on a bench-top centrifuge at top speed for 10 min to remove cells and cell debris. Supernatants and lysates were incubated with the Alkaline Phosphate (AP) substrate p-nitrophenyl phosphate (pNPP) at room temperature. AP activity measured using a 96-well plate spectrophotometer (405 nm). Results are presented as PMA or IO 'shedding over total' calculated by the formula ('cleared stimulated shedding') / ('cleared stimulated shedding' plus' unshed'). 'Cleared stimulated shedding' denotes 'stimulated shedding' minus 'basal shedding'. The release into the medium of a secreted form of luciferase containing a signal peptide was assessed as a control, as described (*Christova et al., 2013*).

### In vitro TACE enzymatic activity assay

The assay was performed as previously described (*Adrain et al., 2012*). In brief, 8 million HEK 293ET cells were lysed for 10 min on ice in 1% Triton X-100, 150 mM NaCl, 50 mM Tris-HCl, pH 7.4 containing complete protease inhibitors cocktail (Roche). Importantly, all steps were performed in the absence of the metalloprotease inhibitor 1,10-phenanthroline to preserve TACE activity. As previously described (*Schlöndorff et al., 2000*) when lysates are made in the absence of 1,10-Phenathroline, TACE autocatalytically cleaves off its cytoplasmic tail, result in loss of the epitope detected by the rabbit polyclonal antibody (Ab39162, Abcam) used for western blotting (*Adrain et al., 2012*). Therefore, mouse anti-TACE (9301, R and D), which recognizes an epitope within the ectodomain, was used for immunoprecipitations, to ensure that TACE was captured regardless of autocatalysis. Mature TACE without its cytoplasmic domain retains proteolytic activity in vitro (*Adrain et al., 2012*). Anti-TACE or mouse IgG (anti-GFP; mock) antibodies were incubated overnight with lysates, followed by capture of the immunocomplexes with anti-mouse magnetic beads. Immunoprecipitates were mixed with the fluorogenic TACE substrate peptide (ENZO Life Sciences, BML-P235-0001) and fluorescence was measured over 3 hr on a Victor three plate reader at 37°C, according to manufacturer instructions. As the western blots used the rabbit polyclonal antibody (Ab39162, Abcam), immunoprecipitates consequently show no evidence of mature TACE.

## Protein extraction from mouse tissues

Protein was extracted from mouse tissues by lysing in a modified RIPA buffer (150 mM NaCl, 50 mM Tris-HCl, pH 7.4, 1 mM EDTA, 1% Triton X-100, 1% Na Deoxycholic Acid, 0.1% SDS containing protease inhibitors and 10 mM 1,10-phenanthroline). Homogenates were clarified and normalized, then incubated with ConA resin, as described above.

## Densitometry

Semi-quantitative densitometric analysis on scanned images from western blot exposures was performed with Fiji software, measuring at least three independent experiments. Results represent mean values ± standard deviation. Half-life was calculated using GraphPad Prism software.

## Alignments

Alignments were performed using Geneious software using the CLUSTALW algorithm.

## Statistical analysis

All statistical analyses were done with two-sample two tail unpaired t-tests assuming unequal variances, using excel software. In *Figure 3*, comparisons were performed after transforming the raw values into their relative (fold change) values to the WT sample. The Pearson's correlation and Mander's colocalization coefficient were calculated after appropriate thresholding with the Volocity (Perkin Elmer) image analysis package. Thresholds were applied evenly across conditions. For the statistical analysis of the Pearson's correlation, the Pearson's taken from at least 20 individual cells acquired over two independent experiments were subjected to unpaired, two tailed t-tests in Excel. P-Values above 0.05 were considered as not significant.

## Acknowledgements

We thank Duarte Barral, Felix Randow, Shigeki Higashiyama, Jürgen Scheller and Feng Zhang for plasmids and Joachim Grötzinger for anti-TACE antibodies. We thank Matthew Freeman for helpful discussions. We thanks Duarte Barral, Maria João Amorim and Claudia Almeida helpful discussions. We express our gratitude to Moises Mallo for advice concerning CRISPR, CRISPR reagents and the generation of iTAP KO mice. We are grateful for the assistance of Ana Nóvoa and IGC's transgenics and mouse facilities. We thank the Life Imaging Center (LIC) of the University of Freiburg for their support regarding the analysis of iRhom2 and lysosomes. We thank IGC's cell sorting/flow cytometry, sequencing, and histopathology facilities and IGC's antibody service (Ana Regalado). We thank Petra Rampírová for cloning and technical assistance. CA acknowledges the support of Fundação Calouste Gulbenkian, Worldwide Cancer Research (14–1289), a Marie Curie Career Integration Grant (project no. 618769), Fundação para a Ciência e Tecnologica (FCT, SFRH/BCC/52507/2014; PTDC/ BEX-BCM/3015/2014; LISBOA-01–0145-FEDER-031330) the European Crohn's and Colitis organization (ECCO), and COST BM1406. SJM acknowledges the support of Science Foundation Ireland (14/ IA/2622). KS acknowledges the support of EMBO (Installation Grant no. 2329), Ministry of Education, Youth and Sports of the Czech Republic (project no. LO1302) and European Regional Development Fund (project OPPK no. CZ.2.16/3.1.00/24016). FS was supported by an Emmy Noether scholarship from the German Research Council, DFG STE2310/1-1. IO acknowledges the support of Fundação Calouste Gulbenkian – IGC (fellowship contract as ref. 91/BD/14). MC acknowledges the support of the FCT (grant SFRH/BPD/117216/2016). This work was developed with the support of the research infrastructure Congento, project LISBOA-01–0145-FEDER-022170, co-financed by Lisboa Regional Operational Programme (Lisboa 2020), under the Portugal 2020 Partnership Agreement, through the European Regional Development Fund (ERDF), and Foundation for Science and Technology (Portugal).

## Additional information

### Funding

| Funder | Grant reference number | Author |
|---|---|---|
| Fundação Calouste Gulbenkian | 91/BD/14 | Ioanna Oikonomidi |
| Fundação para a Ciência e a Tecnologia | SFRH/ BPD/117216/2016 | Miguel Cavadas |
| Science Foundation Ireland | 14/IA/2622 | Seamus J Martin |

| | | |
|---|---|---|
| Deutsche Forschungsgemeinschaft | Emmy Noether scholarship DFG STE2310/1-1 | Florian Steinberg |
| European Molecular Biology Organization | Installation Grant no. 2329 | Kvido Strisovsky |
| Ministerstvo Školství, Mládeže a T?lovýchovy | LO1302 | Kvido Strisovsky |
| European Regional Development Fund | CZ.2.16/3.1.00/24016 | Kvido Strisovsky |
| Worldwide Cancer Research | 14-1289 | Colin Adrain |
| Fundação para a Ciência e a Tecnologia | SFRH/BCC/52507/2014 | Colin Adrain |
| Fundação Calouste Gulbenkian | | Colin Adrain |
| Seventh Framework Programme | Marie Curie Career Integration Grant (project no. 618769 | Colin Adrain |
| The European Crohns and Colitis Organization | | Colin Adrain |
| Fundação para a Ciência e a Tecnologia | PTDC/BEX-BCM/3015/2014 | Colin Adrain |
| European Cooperation in Science and Technology | BM1406 | Colin Adrain |
| Fundação para a Ciência e a Tecnologia | BEX-BCM/3015/2014 | Colin Adrain |
| Fundação para a Ciência e a Tecnologia | LISBOA-01-0145-FEDER-031330 | Colin Adrain |

The funders had no role in study design, data collection and interpretation, or the decision to submit the work for publication.

## Author contributions

Ioanna Oikonomidi, Conceptualization, Data curation, Formal analysis, Investigation, Methodology, Writing—original draft, Project administration, Writing—review and editing; Emma Burbridge, Conceptualization, Investigation, Methodology; Miguel Cavadas, Conceptualization, Investigation; Graeme Sullivan, Heike Naegele, Danielle Clancy, Tianyi Hu, Investigation; Blanka Collis, Alfonso Bolado, Investigation, Methodology; Jana Brezinova, Andrea Bileck, Methodology; Christopher Gerner, Alex von Kriegsheim, Conceptualization; Seamus J Martin, Kvido Strisovsky, Conceptualization, Writing—original draft, Writing—review and editing; Florian Steinberg, Conceptualization, Formal analysis, Investigation; Colin Adrain, Conceptualization, Supervision, Funding acquisition, Methodology, Writing—original draft, Project administration, Writing—review and editing

## Author ORCIDs

Ioanna Oikonomidi http://orcid.org/0000-0003-1657-962X
Emma Burbridge http://orcid.org/0000-0002-4243-7193
Kvido Strisovsky http://orcid.org/0000-0003-3677-0907
Colin Adrain http://orcid.org/0000-0001-7597-4393

## Ethics

Human subjects: Human bloods were obtained from healthy volunteers with informed consent, after review and approval by Trinity College Dublin's research ethics committee.

Animal experimentation: Experiments with mice were performed in accordance with protocols approved by the Ethics Committee of the Instituto Gulbenkian de Ciencia and the Portuguese National Entity (DGAV- Direcao Geral de Alimentacao e Veterinaria) and in accordance with the Portuguese (Decreto-Lei no. 113/2013) and European (directive 2010/63/EU) legislation related to housing, husbandry, and animal welfare.

Decision letter and Author response
Decision letter https://doi.org/10.7554/eLife.35032.034
Author response https://doi.org/10.7554/eLife.35032.035

## Additional files

### Data availability

We have provided the source data for all experiments that involved quantitative analyses.

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
