## [Decision Letter]

Thank you for submitting your article "iTAP, a novel iRhom interactor, controls TNF secretion by policing the stability of iRhom/TACE" for consideration by *eLife*. Your article has been favorably evaluated by Ivan Dikic (Senior Editor) and three reviewers, one of whom, Christopher G Burd (Reviewer #1), is a member of our Board of Reviewing Editors.

The reviewers have discussed the reviews with one another and the Reviewing Editor has drafted this decision to help you prepare a revised submission.

Summary:

This study by Oikonomidi et al. addresses the role of a newly identified protein that associates with iRhom, which the authors call iTAP. The results suggest that iTAP, iRhom2, and TACE form a tripartite complex in cells, and that the elimination of iRhom or iTAP impairs activation of TACE. It is further shown that the stability of these proteins is compromised in the absence of the other subunits and the authors suggest that this is due to lysosome-mediated turnover. Finally, it is shown in mouse and human macrophages that iTAP is necessary for shedding of TNF. Altogether, a strong case is made that iTAP is a component of a TACE complex required for physiological TNF shedding. The trafficking data, particularly those used to argue for lysosomal turnover, are less definitive.

Essential revisions:

1) Direct protein binding interactions are not demonstrated. The text should clearly make this distinction or this should be demonstrated using pure proteins (or fragments thereof).

2) The interpretation of the TACE levels and processing patterns presented in Figure 5 is problematic. The Figure 5B blots of the L929 and RAW cell material show one band that does not appear to co-migrate with the unprocessed or mature forms of TACE that are produced by the HEK and MEF cells (indicted with arrows). The interpretation of the EndoH, F experiments presented in Figure 5C depends critically on this. In panel 5C, the lower band of the EndoF-treated samples is considered the mature form, but no mature form is visible in the untreated lanes. This suggests that TACE is being proteolyzed during the Endo treatments, rather than in cells. Please explain and/or repeat the experiment in a manner that controls for non-physiological proteolysis.

3) The reviewers felt that the conclusion that iTap promotes the 'stability' of TACE, and hence cell surface residence, is not sufficiently supported by the data. It is not apparent from the data shown in Figure 6 that turnover rate of any of the three proteins is reduced by overexpression of iTAP. The cycloheximide chase data shown in Figure 6 needs to be quantified, In Figure 5E, which form of TACE is biotinylated at the cell surface (see point 2)?

Related, throughout the text it is proposed that the proteolytic activity of TACE is affected by loss of iRhom or iTAP, but this is never directly tested and this information is critical for understanding the roles of iTAP and iRhom. The legend from Figure 4 serves as an example: "Figure 4. KO of iTAP diminishes TACE proteolytic activity." The proteolytic activity of surface associated TACE on control versus iTAP (and/or IRhom) null cells should be quantified; is it that the specific activity of cell surface TACE is reduced or that there is simply less TACE on the cell surface? It is noted that there is a commercial TACE enzyme assay using a fluorogenic Peptide Substrate Mca-PLAQAV-Dpa-RSSSR-NH2 (R&D systems).

4) Figure 6E The doublet that is referred to in the text is not apparent. While the endo F treatment does appear to collapse the material to a more prominent band compared to endoH (suggesting that some of the material is endo H resistant), these data are not very compelling. Either the data need to convincingly show the two bands that are referred to, or the experiment should not be included in the manuscript.

5) The conclusion that FRMD8 prevents lysosome-mediated degradation of mature TACE is inferred from the recovery of mature TACE protein level by lysosomotropic agents, but it is never directly demonstrated that TACE (or iRhom) is delivered to the lysosome. Absent more information about sorting of the iRhom/iTAP/TACE complex (does the complex promote recycling to the plasma membrane? limit rate of endocytosis?, etc.), this should be explicitly shown and could be easily addressed by comparing immunofluorescence patterns of TACE and a lysosome resident.

---

## [Author Response]

Essential revisions:1) Direct protein binding interactions are not demonstrated. The text should clearly make this distinction or this should be demonstrated using pure proteins (or fragments thereof).

This is an important point and we have now revised the text to indicate that direct binding was not demonstrated:

“Further studies are required to assess whether the specific binding of iTAP to iRhom is direct, or via an intermediary.”

2) The interpretation of the TACE levels and processing patterns presented in Figure 5 is problematic. The Figure 5B blots of the L929 and RAW cell material show one band that does not appear to co-migrate with the unprocessed or mature forms of TACE that are produced by the HEK and MEF cells (indicted with arrows). The interpretation of the EndoH, F experiments presented in Figure 5C depends critically on this.

We apologize that the resolution in the original Figure 5B made it difficult to discern the immature versus mature species of TACE. Heavily glycosylated proteins, like TACE, are often difficult to separate effectively by electrophoresis. We note that, as for many glycoproteins, differential glycosylation in different cell lines may change the relative mobility of the two TACE species. This is precisely why we used deglycosylation (current Figure 4C, D) to improve the resolution. This is a standard technique often applied to glycoprotein analysis.

Nevertheless, we have now improved the separation of immature/mature TACE by improving the denaturation/reducing conditions (0.5% SDS and 50 mM DTT; increasing the denaturation temperature to 95°C). We believe that the resultant new panel in Figure 4B makes it easier to discern the immature/mature TACE species.

We have added new panels of the deglycosylation experiments (new Figure 4C, D) which we believe resolve the immature vs mature species of TACE much better. Finally, we note that we have used multiple iTAP KO cell lines of different species (human, mouse) all of which reveal the same effect.

In panel 5C, the lower band of the EndoF-treated samples is considered the mature form, but no mature form is visible in the untreated lanes.

A potential explanation is that heterogeneity in the molecular weight of differently glycosylated species of mature TACE results in a diffuse electrophoretic migration of the mature species. This then becomes a much sharper (i.e., more visible) band when deglycosylated with PNGAse F into a single species. We observed this in our previous studies (Adrain et al., 2012). We believe that the resolution is improved in the new panels (new Figure 4 C, D) in the revised manuscript.

This suggests that TACE is being proteolyzed during the Endo treatments, rather than in cells. Please explain and/or repeat the experiment in a manner that controls for non-physiological proteolysis.

We do not believe that this is a consequence of proteolysis. In addition to the comments above, we note that the bands revealed by PNGase F are of precisely the anticipated molecular weight corresponding to deglycosylated TACE minus only its prodomain (Schlondorff, Becherer and Blobel, 2000). Second, the lysis conditions of all of our experiments, pre- and post-revision, included a robust cocktail of protease inhibitors (Aprotinin (2ug/ml, Leupeptin (2 μg/ml), Pepstatin A (1 µg/ml) and AEBSF (80 µM)) in addition to the metalloprotease inhibitor 1-10 phenanthroline (10 mM)(Schlöndorff et al., 2000). Further, the deglycosylation reactions were performed on heat-denatured/reduced protein samples containing 0.5% SDS and 50 mM DTT. To reiterate, we believe that the major issue is the heterogeneity in glycans and the known difficulty in the electrophoretic resolution of heavily glycosylated proteins which can only be accurately resolved by deglycosylation.

3) The reviewers felt that the conclusion that iTap promotes the 'stability' of TACE, and hence cell surface residence, is not sufficiently supported by the data. It is not apparent from the data shown in Figure 6 that turnover rate of any of the three proteins is reduced by overexpression of iTAP. The cycloheximide chase data shown in Figure 6 needs to be quantified.

We agree with the reviewer that this conclusion was not demonstrated sufficiently in the original version. We have now addressed this issue by examining the cell surface stability of iRhom2 (new Figure 5G). We find that the co-expression of iTAP increases the steady state levels of cell surface iRhom2 as well as the stability over a timecourse (new Figure 5G). In addition we have quantitated the experiment in Figure 5B and are now able to conclude that iTAP expression increases the half-life of iRhom2 (from 1.3 hours in the control, to 4.4 hours in cells expressing iTAP).

In Figure 5E, which form of TACE is biotinylated at the cell surface (see point 2)?

This is an important point which we have now clarified fully. We provide additional experiments which indicate, as anticipated from previous studies (Schlondorff, Becherer and Blobel, 2000), that mature TACE is the only form found (biotinylated) at the cell surface. This new data (Figure 4F, right hand panel) is included alongside the original cell surface biotinylation experiment (now Figure 4F, left hand panel). The PNGase F reactions indicate that the only species found on the cell surface of WT cells is the mature TACE. This band is depleted in the iTAP KO cells and is not found on the cell surface of iTAP KO cells.

Related, throughout the text it is proposed that the proteolytic activity of TACE is affected by loss of iRhom or iTAP, but this is never directly tested and this information is critical for understanding the roles of iTAP and iRhom. The legend from Figure 4 serves as an example: "Figure 4. KO of iTAP diminishes TACE proteolytic activity." The proteolytic activity of surface associated TACE on control versus iTAP (and/or IRhom) null cells should be quantified; is it that the specific activity of cell surface TACE is reduced or that there is simply less TACE on the cell surface? It is noted that there is a commercial TACE enzyme assay using a fluorogenic Peptide Substrate Mca-PLAQAV-Dpa-RSSSR-NH2 (R&D systems).

We thank the reviewers for pointing out that we have introduced ambiguity by referring to “proteolytic activity”. We have changed the relevant paragraph headings to read “iTAP-deficient cells are impaired in the shedding of TACE substrates” and “Mature TACE is specifically reduced in iTAP-deficient cells”. We have also amended the text carefully to refer to proteolytic activity only where it is strictly appropriate.

Our data demonstrates (1) that mature TACE (i.e. the proteolytically active form) is specifically depleted in iTAP KO cells (Figure 4) and further (2) that cell surface levels of mature TACE are substantially reduced in iTAP KO cells (Figure 4F). These observations are sufficient to reconcile why the release of TACE substrates is defective, without having to invoke the possibility of altered specific activity.

Unfortunately, it would be extremely difficult to make a side-by-side comparison of the specific activity of equal amounts of mature TACE in WT cells, versus the minority of mature TACE that survives in iTAP KO cells. This would require a specific method to isolate *equal* amounts of *only* the mature species. However, as a useful surrogate, we have now carried out *in vitro* peptide hydrolysis assays to determine the activity of TACE in WT versus iTAP KO cells. These data (Figure 3—figure supplement 1) show that the activity of TACE in immunoprecipitates from iTAP KO cells is, as expected, significantly reduced.

Taken together, we conclude that iTAP KO cells have reduced capacity to release TACE substrates because mature (active) TACE is depleted, as suggested by the reviewer.

4) Figure 6E The doublet that is referred to in the text is not apparent. While the endo F treatment does appear to collapse the material to a more prominent band compared to endoH (suggesting that some of the material is endo H resistant), these data are not very compelling. Either the data need to convincingly show the two bands that are referred to, or the experiment should not be included in the manuscript.

We agree that the deglycosylation data with endogenous iRhom2 is more difficult to resolve, as noted in our previous work (Adrain et al., 2012). However, this panel specifically examines endogenous, not overexpressed, iRhom2 in a setting (macrophage-like cells) highly relevant for iRhom2 biology. It also comes to the same conclusion as our experiment in which overexpressed iRhom2 was analyzed. Rather than removing the data completely, we propose to move this ‘endogenous’ experiment to the supplementary data (Figure 5—figure supplement 1B). The overexpression experiment (new Figure 5F) will be used as the major evidence that iRhom2 leaves the ER in iTap KO cells, with the ‘endogenous’ experiment providing additional reinforcement in the figure supplement. We hope that this is a suitable solution.

5) The conclusion that FRMD8 prevents lysosome-mediated degradation of mature TACE is inferred from the recovery of mature TACE protein level by lysosomotropic agents, but it is never directly demonstrated that TACE (or iRhom) is delivered to the lysosome. Absent more information about sorting of the iRhom/iTAP/TACE complex (does the complex promote recycling to the plasma membrane? limit rate of endocytosis?, etc.), this should be explicitly shown and could be easily addressed by comparing immunofluorescence patterns of TACE and a lysosome resident.

We agree that the paper needed strengthening in these aspects and thank the reviewer for the suggestions. We have now added several pieces of new data to address these issues:

1) As suggested by the reviewer, we now show using fluorescence microscopy that in iTAP KO cells, iRhom2 is delivered to the lysosome (Figure 6A). This defect is rescued by re-expressing iTAP in the KO cells (Figure 6B). This is consistent with our original biochemical experiments that used lysosomotropic drugs to reach the same conclusion.

2) In a gain-of-function model we show that overexpression of TACE alone does not result in its recruitment to the lysosome, presumably because it is not present at stoichiometric ratios with iRhom2. By contrast, over-expressed iRhom2 leads to its recruitment to the lysosome (Figure 6E) and, in turn, recruits TACE to the lysosome too. This suggests that, in the absence of normal iTAP stoichiometry to iRhom2, the sheddase complex (e.g. TACE, iRhom2) is subject to a default trafficking itinerary to the lysosome.

3) Further, we show that iTAP does not colocalize with early endosomes (Figure 8—figure supplement 1D), implying that it is not a direct component of the endocytic trafficking machinery and is hence unlikely to directly promote recycling or directly impinge on endosomes.

4) In addition, we show that iTAP expression increases the steady state levels of cell surface iRhom2 as well as the cell surface stability of the protein (Figure 5G).

In summary, relating the role of iTAP to the wider themes amongst FERM domain biology, we can conclude that iTAP does not form an intimate connection with the endocytic recycling machinery. Instead, iTAP’s apparent role appears to more closely resemble the canonical function of FERM domain proteins in stabilizing (anchoring) membrane proteins on the cell surface.